# SMN-deficient cells exhibit increased ribosomal DNA damage

Evangelia Karyka[1,3], Nelly Berrueta Ramirez[1,2], Christopher P Webster[1,3], Paolo M Marchi[1,3], Emily J Graves[1,3], Vinay K Godena[3], Lara Marrone[3], Anushka Bhargava[3], Swagat Ray[2,5], Ke Ning[1,3], Hannah Crane[2], Guillaume M Hautbergue[1,3], Sherif F El-Khamisy[1,2,4,*], Mimoun Azzouz[1,3,*]

**Spinal muscular atrophy, the leading genetic cause of infant mortality, is a motor neuron disease caused by low levels of survival motor neuron (SMN) protein. SMN is a multifunctional protein that is implicated in numerous cytoplasmic and nuclear processes. Recently, increasing attention is being paid to the role of SMN in the maintenance of DNA integrity. DNA damage and genome instability have been linked to a range of neurodegenerative diseases. The ribosomal DNA (rDNA) represents a particularly unstable locus undergoing frequent breakage. Instability in rDNA has been associated with cancer, premature ageing syndromes, and a number of neurodegenerative disorders. Here, we report that SMN-deficient cells exhibit increased rDNA damage leading to impaired ribosomal RNA synthesis and translation. We also unravel an interaction between SMN and RNA polymerase I. Moreover, we uncover an spinal muscular atrophy motor neuron-specific deficiency of DDX21 protein, which is required for resolving R-loops in the nucleolus. Taken together, our findings suggest a new role of SMN in rDNA integrity.**

## Introduction

The nucleolus is a dynamic nuclear membrane-less organelle in which ribosomal DNA (rDNA) transcription and ribosomal assembly take place. rDNA is transcribed by RNA polymerase I in a cell cycle phase-dependent manner (Sirri et al, 2008). The size and number of nucleoli per cell depend on the rate of RNA polymerase I–mediated transcription, which in turn, is influenced by cell growth and metabolic state (Russell & Zomerdijk, 2005). Importantly, the nucleolus is not only the ribosome factory of a cell; it is also a stress sensor, whose activity is modulated by stress conditions (Olson, 2004; Pederson & Tsai, 2009; Boulon et al, 2010).

Nucleolar disruption could be caused by increased rDNA damage among other factors (van Sluis & McStay, 2017). The repetitive nature of rDNA coupled with its high transcription rates could lead to improper recombination and potential rDNA deletions or rearrangements as well as to the formation of RNA/DNA hybrid structures known as R-loops, composed of a displaced single-stranded DNA hybridized with the nascent transcript (Santos-Pereira & Aguilera, 2015). Cells have developed several mechanisms to degrade or unwind R-loops, in particular these structures can be resolved by degradation of the RNA moiety in the RNA/DNA hybrid mediated by endonucleases such as RNAse H enzymes (Wahba et al, 2011). Alternatively, R-loops can be unwound by members of the helicase family, such as senataxin, Pif1, and the DEAD-box helicases DHX9, DDX19, DDX23, and DDX21 (Chakraborty & Grosse, 2011; Skourti-Stathaki et al, 2011; Hodroj et al, 2017; Song et al, 2017; Sridhara et al, 2017; Tran et al, 2017). Accumulation of nucleolar R-loops leads to rDNA double-strand breaks (DSBs) (Tsekrekou et al, 2017). DSBs are the most deleterious type of DNA damage (Pankotai & Soutoglou, 2013). rDNA DSBs have been shown to result in an ataxia-telangiectasia mutated-dependent inhibition of RNA polymerase I transcription (Kruhlak et al, 2007) and nucleolar reorganization with the formation of nucleolar caps (Harding et al, 2015). Perturbations in any step of ribosomal biogenesis such as rDNA transcription, ribosomal RNA (rRNA) processing and ribosomal assembly can cause nucleolar disruption and p53-mediated cell cycle arrest or apoptosis (Rubbi & Milner, 2003; James et al, 2014). Therefore, it is not surprising that normal nucleolar function is critical for cell survival.

Of note, nucleolar stress has been observed in several neurodegenerative diseases, such as Parkinson's disease, Huntington's disease, Alzheimer's disease, and Amyotrophic Lateral Sclerosis, suggesting this might be a common denominator in neurodegeneration (Iacono et al, 2008; Rieker et al, 2011; Lee et al, 2014; Parlato & Liss, 2014).

[1]The Healthy Lifespan Institute and Neuroscience Institute, Neurodegeneration and Genome Stability Group, University of Sheffield, Sheffield, UK  [2]Department of Molecular Biology and Biotechnology, The Institute of Neuroscience and the Healthy Lifespan Institute, School of Bioscience, Firth Court, University of Sheffield, Sheffield, UK  [3]Sheffield Institute for Translational Neuroscience, University of Sheffield, Sheffield, UK  [4]The Institute of Cancer Therapeutics, University of Bradford, Bradford, UK  [5]Department of Life Sciences, School of Life and Environmental Sciences, University of Lincoln, Lincoln, UK

Correspondence: m.azzouz@sheffield.ac.uk; s.el-khamisy@sheffield.ac.uk
Sherif F El-Khamisy and Mimoun Azzouz are joint senior authors.
*Sherif F El-Khamisy and Mimoun Azzouz contributed equally to this work.

Spinal muscular atrophy (SMA) is a fatal autosomal recessive neurodegenerative disorder characterised by selective loss of *α* motor neurons in the anterior horn of the spinal cord leading to muscle atrophy and weakness (Wang et al, 2007). It is considered the most common genetically inherited neurological disorder resulting in infant mortality (Nash et al, 2016). SMA is caused by homozygous mutations or deletion of the *Survival Motor Neuron 1* gene (*SMN1*). This loss of function leads to reduced levels of the ubiquitously expressed SMN protein (Kolb & Kissel, 2011). SMN is a multifunctional protein and it is still unclear which of the numerous functions of SMN is essential for motor neuron survival. Motor neurons typically have very prominent nucleoli because of their strong energy demand, which requires high levels of ribosome synthesis (Lafarga et al, 1991; Berciano et al, 2007; Jordan et al, 2007; Le Masson et al, 2014). Emerging evidence suggests that SMN may have a role as a guardian of genome integrity (Zhao et al, 2016; Jangi et al, 2017; Kannan et al, 2018, 2020). Interestingly, DNA damage and genome instability have been linked to numerous neurodegenerative diseases (McKinnon, 2009; Jeppesen et al, 2011; Madabhushi et al, 2014; Carroll et al, 2015; Walker et al, 2017). The rDNA represents a particularly unstable locus undergoing frequent breakage (Marnef et al, 2019), and rDNA instability has been linked to neurodegenerative disorders among others (Killen et al, 2009; Stults et al, 2009; Hallgren et al, 2014).

Here, we present that SMN-deficient cells exhibit increased rDNA damage and, nucleolar reorganization with the formation of nucleolar caps. This nucleolar disruption leads to impaired rRNA synthesis and translation. We also found an SMA motor neuron-specific deficiency of DDX21 protein, which is required for resolving R-loops in the nucleolus. Finally, we uncovered a novel interaction between SMN and RNA polymerase I. Taken together, our findings suggest that increased rDNA damage may contribute to SMA pathogenesis and SMN may play a role in rDNA integrity.

# Results

## SMN-deficient cells exhibit increased nucleolar disruption associated with increased rDNA damage

A growing body of evidence has reported increased incidence of R-loops in SMN-deficient cells (Zhao et al, 2016; Jangi et al, 2017; Kannan et al, 2018, 2020), which we confirmed by labelling SMN-deficient motor neurons with an R-loop specific antibody (S9.6 antibody). When compared with control samples, E13 embryonic spinal motor neurons (Figs 1A and B and S1A and B), spinal cord sections derived from SMA pre-symptomatic mice (postnatal day 2) (Fig S2), postmortem spinal cord tissue derived from SMA type I patients (Fig 1C and D) as well as iPSC-derived motor neurons from SMA type I patients (Fig S3A and B) all displayed increased numbers of R-loops. In non-neuronal cell types, such as fibroblasts, the morphology of R-loop–enriched nuclear structures was also abnormal (Fig 1E and F). Signal specificity was confirmed by pre-treatment with ribonuclease H (RNAse H) enzyme (Figs 1C and S4). RNase H acts by specifically degrading the RNA moiety of RNA/DNA hybrids (Hausen & Stein, 1970; Berkower et al, 1973). The results

obtained through S9.6 staining were further backed-up by orthogonal assays such as HB-GFP ChIP-qPCR, as it will be described below. Given that R-loops are specifically formed at highly transcribed regions such as R-loop prone rDNA arrays (El Hage et al, 2010), it was hypothesized that the observed R-loop staining is primarily nucleolar and that the abnormal phenotype of R-loops in SMN-deficient cells could be due to nucleolar disruption. To test this hypothesis SMA type I fibroblasts and healthy controls were stained with nucleolin, a major nucleolar protein of the dense fibrillar component (DFC) (Tajrishi et al, 2011). As expected, nucleolin staining covered a larger area of the nucleolus in SMA type I fibroblasts than in controls, confirming changes in nucleolar morphology of SMA type I fibroblasts (Fig 1G and H). In SMA cells, adjacent nucleoli seem to have fused or collapsed forming altered, larger nucleolar structures. Similar phenotype was observed in spinal motor neurons isolated from SMNΔ7 mouse embryos, an established and widely used SMA model (Fig 1I and J). A double staining of motor neurons with S9.6 and nucleolin antibodies confirmed the nucleolar localisation of the R-loops seen in SMN-deficient neurons (Fig S5). The SMN levels of all the SMA experimental models used are presented in Figs S6A–D and S7B and C. The generation and characterization of iPSC-derived motor neurons is presented in Fig S7A and D.

Intriguingly, it has been reported that reorganization of nucleolar architecture may be caused by persistent rDNA damage (Harding et al, 2015; van Sluis & McStay, 2015). Hence, we hypothesized that nucleolar abnormalities may be linked to alterations in rDNA integrity. Of note, whereas DNA damage has been largely presented as a contributor of SMA pathogenesis, all research has been focused on nucleoplasmic DNA integrity rather than rDNA integrity in SMN-deficient cells (Zhao et al, 2016; Jangi et al, 2017; Kannan et al, 2018, 2020).

The nucleolus consists of three distinct components: a fibrillary centre (FC), surrounding DFC, and a granular component surrounding the DFC. These three layers exhibit liquid-like properties and their distinct organization is a consequence of liquid–liquid demixing (Feric et al, 2016). Pre-rRNA synthesis takes place in the interphase between FC and DFC, whereas pre-rRNA processing begins in DFC, only to be completed in the granular component, where the ribosomal assembly also occurs (Hinsby et al, 2006; Boisvert et al, 2007). Persistent DSBs in rDNA cause the FCs and DFCs to migrate along with the rDNA to the periphery of the nucleolus, forming nucleolar caps (van Sluis & McStay, 2015). Strikingly, we found a similar formation of nucleolar caps in SMA type I fibroblasts stained with nucleolin. These structures resemble cap-like formations located on the outer part of the segregated nucleolus (Fig 2A, arrows). Notably, these nucleolar caps are associated with γH2AX foci (Fig 2A), a marker of DSBs (Rogakou et al, 1999; Modesti & Kanaar, 2001), suggesting that the rDNA exposed to the periphery is severely damaged. Indeed, when counting the number of nucleolin-positive γH2AX foci per cell, we uncovered significantly increased numbers in SMA type I fibroblasts compared with controls (Fig 2B).

To validate the presence of DSBs in rDNA, we carried out γH2AX–chromatin immunoprecipitation (γH2AX-ChIP), followed by q-PCR analysis of ribosomal genes (*RPL32*, *18S*, *5.8S*, and *28S*), which were all found to be enriched in γH2AX (Fig 2C). Of note, RPL32 (ribosomal protein L32) is a ribosomal protein whose gene is

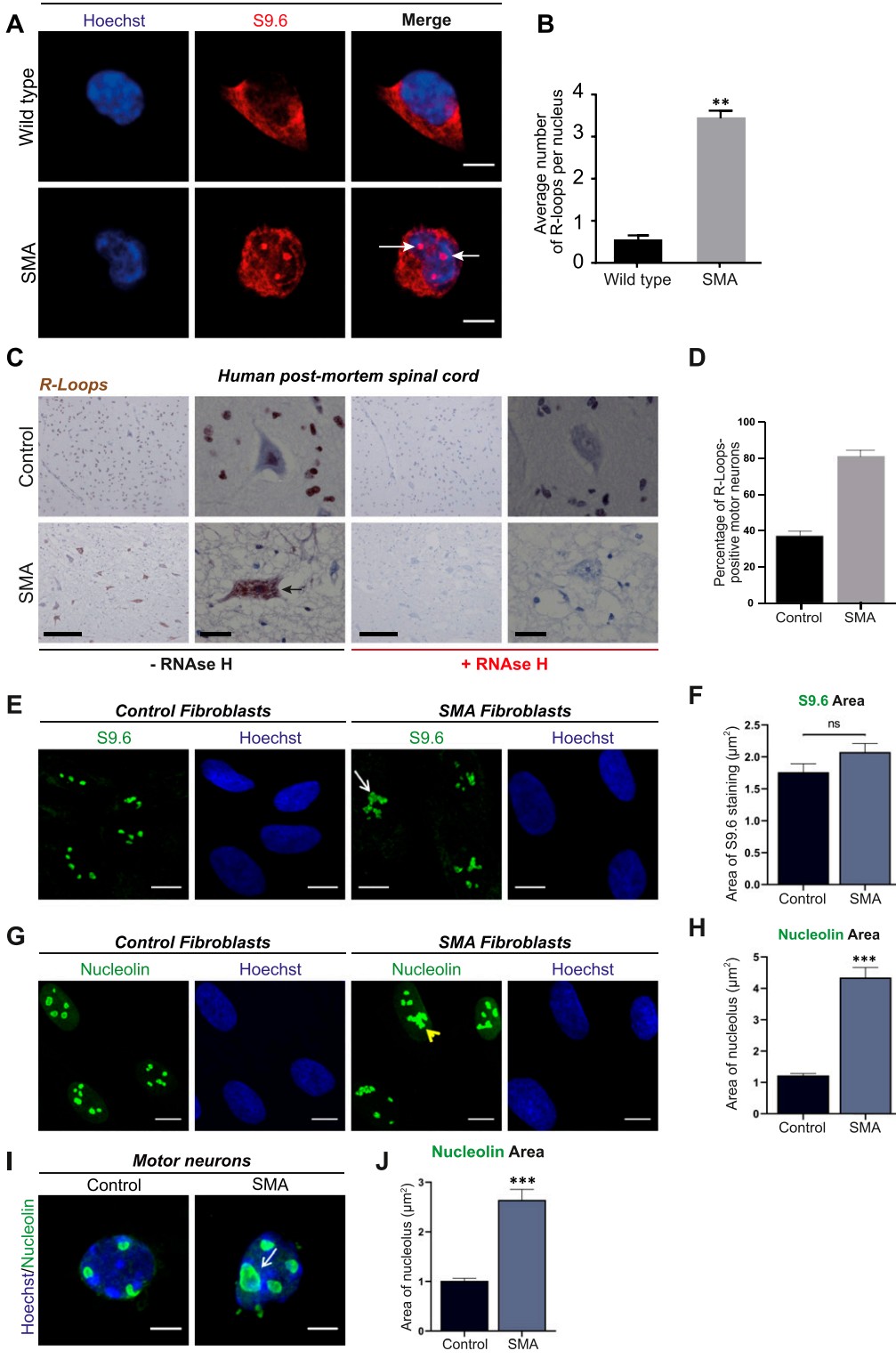

**Figure 1. Accumulation of R loops in survival motor neuron (SMN)–deficient cells and ribosomal disruption.**
**(A)** p75-enriched motor neurons derived from SMNΔ7 and wild type E13 embryos were labelled with RNA/DNA heteroduplex-specific antibody, S9.6 (Kerafast, ENH001) at DIV7. Scale bars represent 5 µm. **(B)** Quantification of the number of RNA/DNA hybrids (R-Loops) in spinal muscular atrophy (SMA) and wild type motor neurons. Data presented as mean ± SEM **$P < 0.01$; paired two-tailed $t$ test ($P = 0.0016$). The data were collected from three biological independent replicates (N = 3). Nuclei counted = 10/replicate. **(C)** Sections of postmortem spinal cord from SMA patients (N = 4) and control individuals (N = 2) were stained for S9.6. Specificity of S9.6 staining confirmed by treating sections with RNAse H enzyme before R-loop labelling. Scale bars represent 50 µm (left images) and 10 µm (right images), respectively. **(D)** Percentage of R-Loop–positive motor neurons in spinal cord sections from SMA patients and control individuals. $P$-value analysis was not performed as only two controls were analysed because of the difficulty in having postmortem tissue from healthy young children. **(E)** Fibroblasts derived from SMA type I child (GM08318) and healthy control (GM00498) were labelled for RNA/DNA hybrids (S9.6). Scale bars represent 10 µm. Arrow indicates an abnormal structure. **(F)** Total area of staining per cell (µm$^2$), determined by boundaries of S9.6 immunofluorescence was measured for control and SMA fibroblasts. Bar graphs of mean ± SEM (N = 3). Mann–Whitney non-parametric test. ns, not significant ($P > 0.05$). **(G)** Fibroblasts derived from SMA type I child and healthy control were stained with nucleolin (green). Yellow arrowhead shows disrupted nucleoli. Scale bars represent 10 µm. **(H)** Total nucleolar area per cell (µm$^2$) determined by boundaries of nucleolin immunofluorescence was measured for Control and SMA fibroblasts. Bar graphs of mean ± SEM (N = 3). Mann–Whitney non-parametric test. ***$P \leq 0.001$. **(I)** E13 motor neurons derived from SMA and wild type mouse embryos were also stained with nucleolin (green) antibody and Hoechst (blue). White arrow shows an enlarged (disrupted) nucleolus. Scale bars represent 5 µm. **(J)** Total nucleolar area per cell (µm$^2$) was measured and data presented as mean ± SEM (N = 3). Mann–Whitney non-parametric test. ***$P \leq 0.001$.

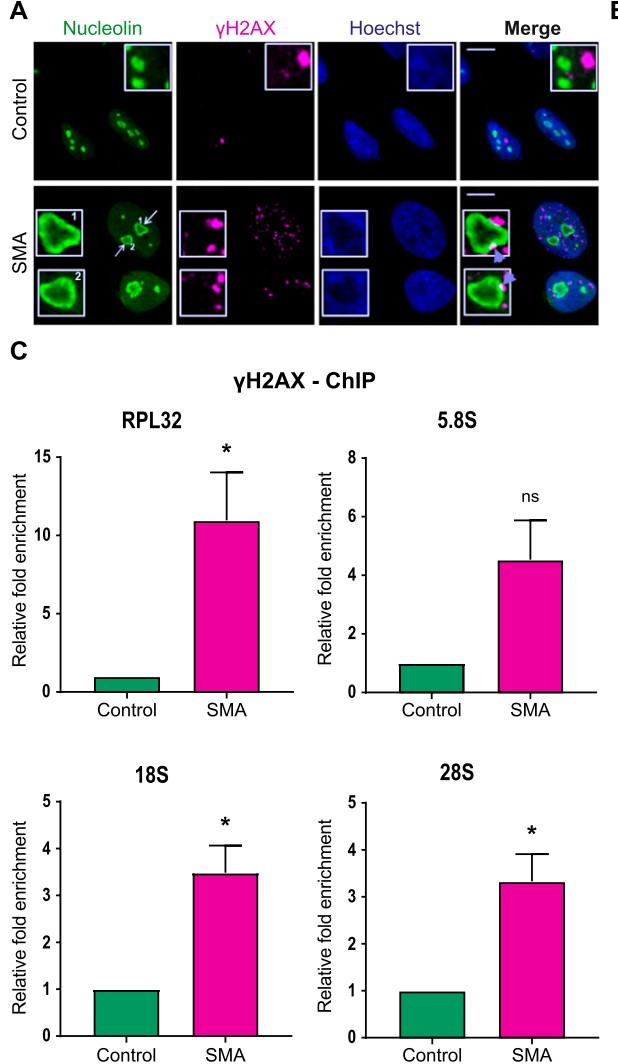

**C**

**γH2AX - ChIP**

**RPL32**

**5.8S**

**18S**

**28S**

**Figure 2. Survival motor neuron–deficient cells exhibit increased ribosomal DNA damage.**
**(A)** Dual immunostaining with nucleolin and γH2AX performed on spinal muscular atrophy (SMA) type I and control fibroblasts. SMA type I fibroblasts form nucleolar caps (white arrows) that are shown to co-localise with γH2AX foci (violet arrowheads). Scale bars represent 5 μm. **(B)** Average number of nucleolin-positive γH2AX foci per nucleus. Data presented as mean ± SEM *$P < 0.05$, paired two-tailed $t$ test ($P = 0.0176$). The data were collected from three biological independent replicates (N = 3). Nuclei counted = 50/replicate. **(C)** γH2AX-ChIP followed by qPCR analysis of ribosomal genes. Quantified RPL32, 5.8S, 18S and 28S gene qPCR data from γH2AX-ChIP experiment in SMA type I fibroblasts and healthy controls. IgG was used as a background control. Bar graphs of mean ± SEM (N = 3). *$P < 0.05$, ns, not significant ($P > 0.05$). Paired two-tailed $t$ test; $P = 0.0472$ (RPL32), $P = 0.0778$ (5.8S), $P = 0.0237$ (18S), $P = 0.0278$ (28S).

transcribed by RNA polymerase II, whereas R NA polymerase I is responsible for the transcription of *18S*, *5.8S* and *28S* genes.

To reinforce our hypothesis that the rDNA damage observed in SMA cells is R-loop mediated, we performed a GFP ChIP-qPCR using a construct that contained the Hybrid binding domain from RNase H1 fused to EGFP (HB-GFP) (Fig 3A). This domain identifies and binds to R-loops with a specificity fivefold higher than the S9.6 antibody (Nowotny et al, 2008). The construct was inserted into SMA type I fibroblasts and controls via lentiviral transduction with an MOI of 30. RNA/DNA hybrids bound to the fusion protein were pulled down; the DNA was purified and the regions amplified by qPCR corresponded to three rDNA sequences 5.8S, 28S, R7, and the nuclear sequence Actin 5′ pause. The R7 sequence encompasses the end of *5.8S* gene and the first part of the internal transcribed spacer 2 (Shen et al, 2017); Actin 5′ pause primers correspond to the 5′ of 3′ pause site (transcription termination site), a region prone to R-loop formation, transcribed by RNA pol II (Skourti-Stathaki et al, 2011). All sequences showed a trend in the increase of R-loops in SMA cells, compared with controls in both the inputs, and the fold enrichment

(Fig 3B and C). Taken together, these data support the idea that SMN deficiency leads to nucleolar disruption and increased R-loop–mediated rDNA damage.

### SMN interacts with RNA polymerase I

Having established that SMN-deficient cells exhibit abnormal R-loops in the nucleolus and increased rDNA, and given that SMN is reported to localise in the nucleolus (Francis et al, 1998), we next hypothesised that there might be a role for SMN in facilitating the resolution of nucleolar R-loops during RNA polymerase I–mediated transcription and preventing DNA instability similarly to what occurs in the nucleoplasm. To confirm our hypothesis we performed immunoprecipitation (IP) experiments using an antibody against RNA polymerase I. Because SMN binds to RNA polymerase II and senataxin in the nucleoplasm to facilitate the resolution of naturally occurring R-loops (Zhao et al, 2016), RNA polymerase II was included as a positive control, whereas rabbit IgG was our negative control. Our results revealed the existence of a novel complex

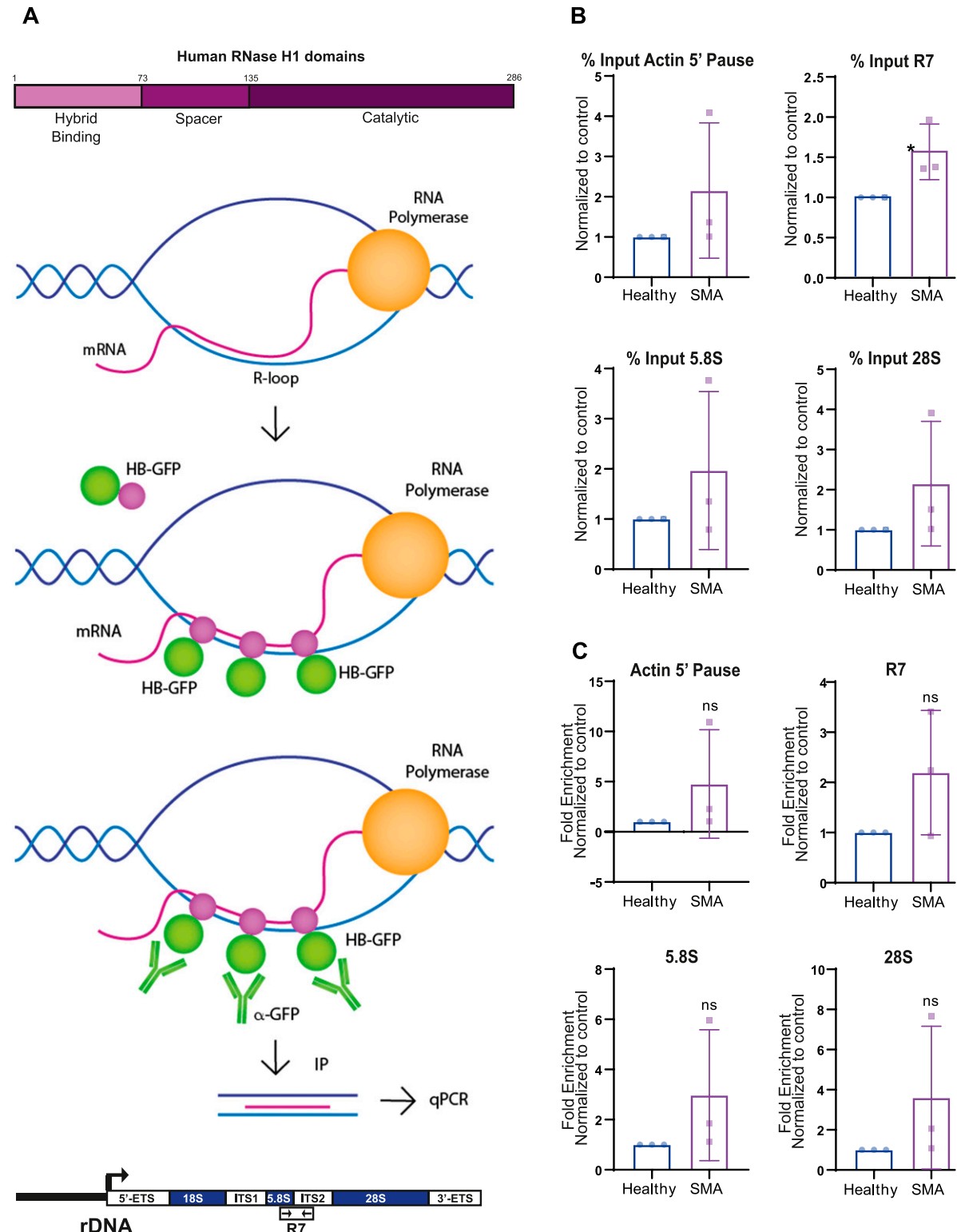

**Figure 3. The ribosomal DNA damage seen in spinal muscular atrophy is R-loop mediated.**
**(A)** Schematic representation of the HB-GFP-ChIP approach. Hybrid binding domain from RNase H1 fused with EGFP detected and bound to DNA/RNA hybrids. R-loops were isolated by pulling down the HB-GFP fusion protein with GFP-trap beads. qPCR-amplified Actin 5′ pause nuclear region; and R7, 5.8S, and 28S ribosomal sequences. **(B, C)** Quantified qPCR data from HB-GFP-ChIP experiment in spinal muscular atrophy type I fibroblasts and healthy controls. Data are presented as mean ± s.d. (N = 3). *P < 0.05, ns, not significant (P > 0.05). **(B)** Percent input, unpaired two-tailed t test; P = 0.3004 (Actin 5′ pause), P = 0.0464 (R7), P = 0.3481 (5.8S), P = 0.2691 (28S). **(C)** Fold enrichment, unpaired two tailed t test; P = 0.2947 (Actin 5′ pause), P = 0.1702 (R7), P = 0.2603 (5.8S), P = 0.2734 (28S).

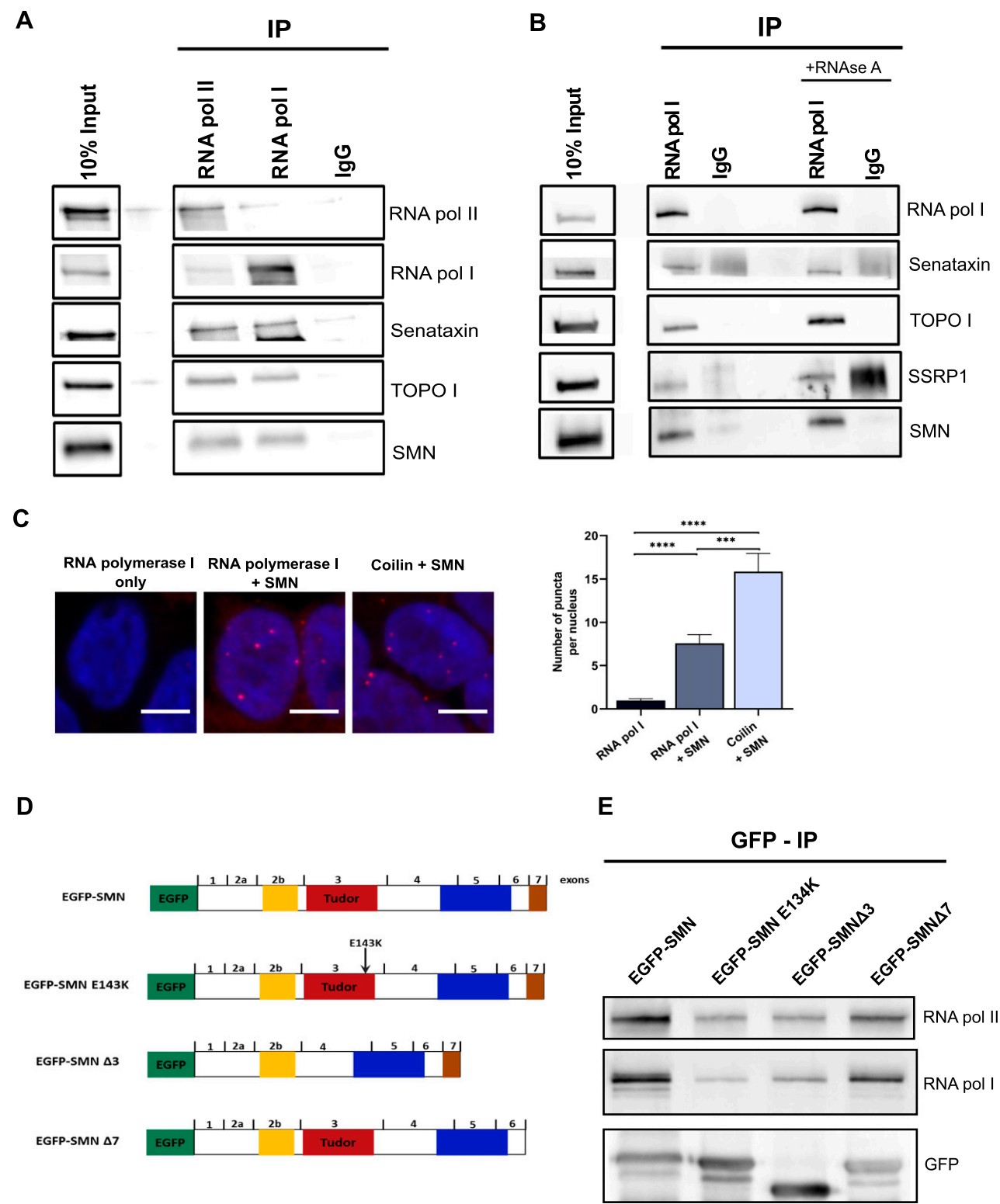

**Figure 4. RNA polymerase I as a novel survival motor neuron (SMN)–interacting protein.**
**(A)** SMN protein interacts with RNA polymerase I. Immunoprecipitation (IP) experiments were carried out in HEK293T cell nuclear extract using anti-RNA polymerase II, anti-RNA polymerase I, or control rabbit (IgG) antibodies. Immunoprecipitates were analysed by SDS–PAGE and Western blotting with antibodies against RNA polymerase II, RNA polymerase I, SMN, and senataxin. **(B)** SMN–RNA polymerase I interaction is RNA independent. Immunoblot analyses of RNA polymerase I, senataxin, SSRP1, and SMN on immunoprecipitations with RNA polymerase I (RNA polymerase I IP) from nuclear extracts of HEK293T cells without and with RNAse A treatment, respectively. Rabbit IgG was used as negative control. **(C)** In situ proximity ligation assay detection of the interaction between SMN and RNA polymerase I in HEK293T cells. Scale bars represent

encompassing RNA polymerase I, SMN, and senataxin (Fig 4A). Incubation of cell extracts with RNAse A further demonstrated that the SMN–RNA polymerase I interaction is RNA-independent (Fig 4B). The SSRP1 (structure specific recognition protein 1) protein, known to be associated with RNA polymerase I (Birch et al, 2009), was used as a positive control. To confirm this interaction in situ, we performed proximity ligation assay (PLA) between endogenous SMN and RNA polymerase I. Nuclear PLA signals were observed indicating an interaction between SMN and RNA polymerase I in intact cells. Abundant PLA puncta (red dots) were detected when SMN and RNA polymerase I antibodies were used. As a positive control, we used SMN and coilin antibodies because coilin is a well-known SMN interactor. Finally, very few puncta were observed when RNA polymerase I antibody was used on its own as a negative control (Fig 4C). Taken together, these studies unravelled a novel interaction between SMN with RNA polymerase I.

We next sought to determine which of the SMN domains would be critical for this interaction. We began by analysing the SMN Tudor domain, a conserved structural motif originally identified as a region of 50 amino acids in *Drosophila*'s Tudor protein. The Tudor domain in human SMN is encoded by exon 3 and presents as a strongly bent anti-parallel β-sheet recognising symmetrically dimethylated arginine residues (Zhao et al, 2016). To evaluate the role of the SMN Tudor domain in the interaction between SMN and RNA polymerase I, we expressed EGFP–SMN, EGFP–SMN E134K, EGFP–SMNΔ3, and EGFP–SMNΔ7 fusion proteins (Figs 4D and S8) in HEK293T cells, followed by co-immunoprecipitation experiments. We discovered that the Tudor domain is required for SMN to bind to RNA polymerase I because the SMNΔ3 construct (which lacks the Tudor domain) had a lower interaction with RNA polymerase I than the full length SMN construct (Figs 4E and S9). Interestingly, an SMA-causing point mutation (E134K) (Selenko et al, 2001) within the Tudor domain in the vicinity of the methyl-arginine binding cage that does not disrupt the Tudor domain but inhibits its binding to the methylated targets showed similar reduction of SMN association with RNA polymerase I (Fig 4E). This observation may be of great significance because SMN has also been shown to interact with RNA polymerase II through its Tudor domain in a methylation-dependent manner (Zhao et al, 2016).

### rDNA damage is associated with impaired rRNA synthesis and translation in SMA cells

A number of studies have shown that increased rDNA DSBs induce nucleolar reorganization with cap formation and inhibition of RNA polymerase I–mediated transcription (Harding et al, 2015; van Sluis & McStay, 2015; Warmerdam et al, 2016). Maintaining appropriate levels of rDNA transcription is crucial for cellular homeostasis because its deregulation can result in human diseases (Russell & Zomerdijk, 2005; Diesch et al, 2014). Interestingly, inhibition of RNA polymerase I–driven transcription is sufficient to induce neuronal

death (Kalita et al, 2008; Parlato et al, 2008; Rieker et al, 2011; Kreiner et al, 2013). For instance, impaired rDNA transcription has been linked to Huntington's disease, Parkinson's disease, Alzheimer's disease, and amyotrophic lateral sclerosis (Lee et al, 2011; Kwon et al, 2014; Garcia-Esparcia et al, 2015; Kang & Shin, 2015; Maina et al, 2018). Because we detected increased rDNA DSBs and cap formation in SMN-deficient cells, we hypothesized that as a result the transcription of rDNA could be inhibited. To investigate whether RNA polymerase I–mediated transcription is also affected in SMN-deficient cells, total RNA from SMA and healthy embryonic spinal motor neurons was isolated and assessed by qRT-PCR to investigate the levels of the 45S precursor as well as the 18S, 5.8S, and 28S mature rRNAs. The levels of all mature rRNA species appeared to be lower in SMA motor neurons compared with controls with the levels of 18S and 5.8S being significantly reduced (Fig 5A). This reduction in rRNA levels suggests impairment in rRNA biogenesis caused, possibly and at least in part, by the increased rDNA damage observed in SMA. The reduced levels of rRNA may then lead to a decrease in the ribosome pool of the cell. Indeed, the results of two independent studies are in agreement with this hypothesis, as reduced ribosomal content was observed in SMA motor neurons (Bernabo et al, 2017; Tapia et al, 2017). Intriguingly, this phenotype appeared to be motor neuron specific, as we did not detect any significant differences in rRNA levels between control and SMA cells when analysing embryonic cortical neurons (Fig 5B).

Hence, our data suggest that increased rDNA instability in SMN-deficient cells is accompanied by poor ribosome biogenesis, particularly in motor neurons.

Because ribosomes carry out cellular translation, we hypothesized that impaired ribosomal biogenesis could lead to defective protein synthesis. To test our hypothesis, we cultured primary motor neurons and measured de novo protein synthesis levels using an established metabolic labelling technique based on incorporating the methionine-homologue L-azidohomoalanine (AHA) into newly synthesized proteins. Similar to what reported by Bernabo et al (2017) we observed a significant decrease in fluorescence intensity in SMN-deficient motor neurons (Fig 5C and D). Intriguingly, virus-mediated overexpression of SMN or SETX reversed this phenotype (Fig 5C and D), suggesting that this translational defect could be corrected by resolving R-loops.

To corroborate our finding that R-loop–mediated rDNA damage is at the core of translational defects, we treated wild type (control) motor neurons with camptothecin (CPT), an anticancer drug that is a known R-loop activator. CPT induces R-loop–dependent DSBs in the absence of DNA replication (Cristini et al, 2019), specifically in highly transcribed nucleolar regions, such as ribosomal genes. Long exposure of motor neurons to low concentrations of CPT (50 nM) resulted in visibly increased rDNA damage. When measuring the intensity of nucleolin-positive γH2AX foci per cell, we observed a significant increase in CPT-treated cells compared with untreated (Fig 6A and B). Notably, protein synthesis was also reduced in

5 µm. Quantification of the number of puncta per nucleus was performed. Bar graphs of mean ± SEM (N = 3). Kruskal–Wallis non-parametric test with Dunn's multiple comparisons. ***P ≤ 0.001, and ****P ≤ 0.0001. **(D)** Schematic representation of EGFP-tagged SMN construct used in the experiments. **(E)** IPs were performed with nuclear extracts from HEK293T cells transiently transfected with EGFP-tagged SMN full length, SMN E134K, SMN Δ3, and SMN Δ7, respectively. The anti-GFP incubation served as a loading control.
Source data are available for this figure.

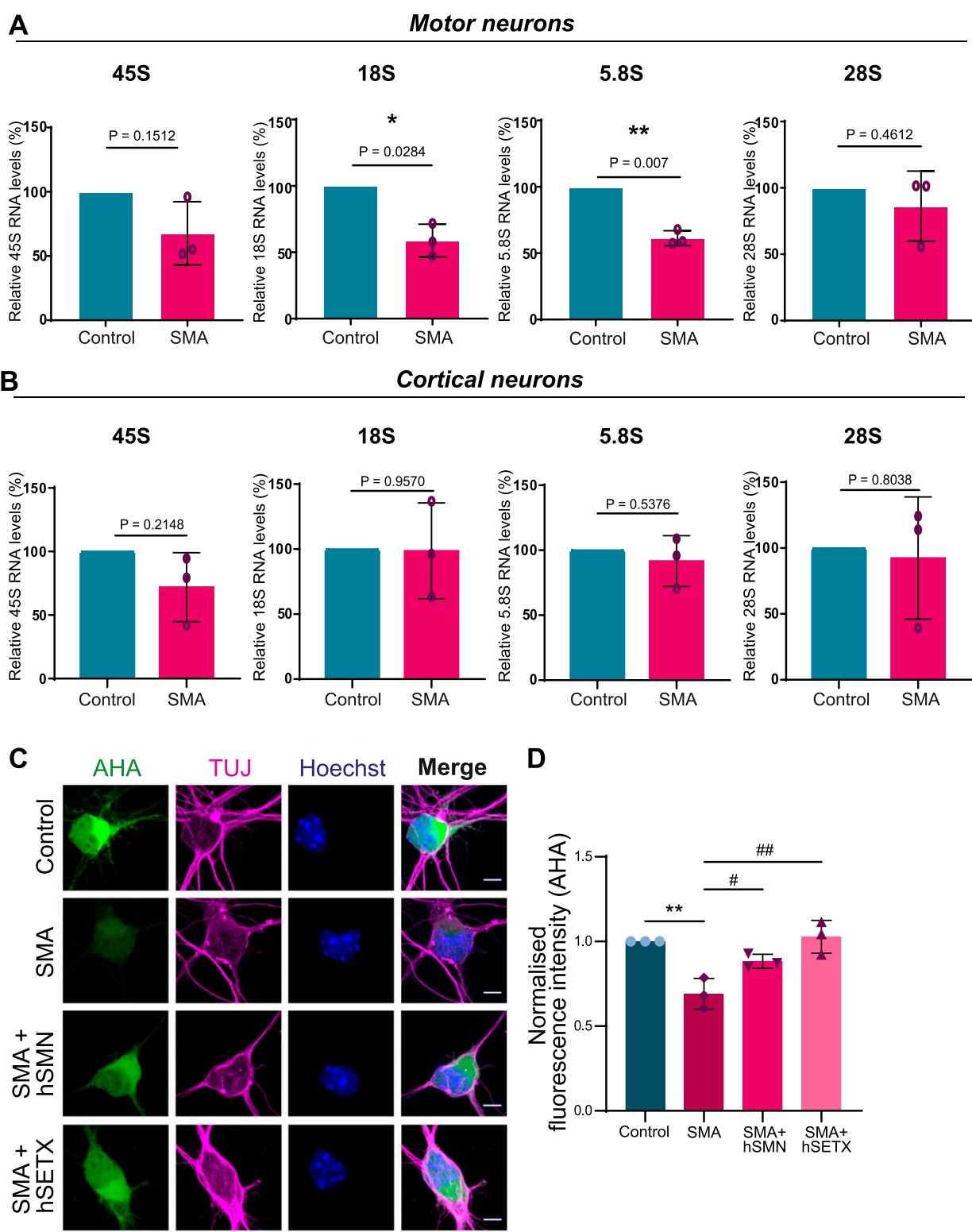

**Figure 5.  Ribosomal DNA damage leads to impaired ribosomal RNA (rRNA) synthesis and translation.**
**(A)** Analysis of rRNA synthesis. Total RNA was extracted from spinal muscular atrophy (SMA) and control embryonic motor neurons. The levels of 45S pre-rRNA along with 18S, 5.8S, and 28S mature rRNAs were determined by quantitative reverse transcription PCR (qRT-PCR) and normalised to GAPDH levels. Data are presented as mean ± SEM (N = 3). *P < 0.05, **P < 0.01 paired two-tailed *t* test; P = 0.1512 (45S), P = 0.0284 (18S), P = 0.007 (5.8S), P = 0.4612 (28S). **(B)** Total RNA was extracted from SMA and control embryonic cortical neurons. The levels of 45S pre-rRNA along with 18S, 5.8S, and 28S mature rRNAs were determined by qRT-PCR and normalised to GAPDH levels. Data are presented as mean ± SEM. (N = 3). Paired two-tailed *t* test; P = 0.2148 (45S), P = 0.9570 (18S), P = 0.5376 (5.8S), P = 0.8083 (28S). **(C)** SMA and control embryonic motor

CPT-treated cells (Fig 6C and D), suggesting that R-loop–mediated rDNA damage can result in translational defects.

## DDX21 deficiency in SMA spinal motor neurons may exacerbate the ribosomal defects

As discussed above, the formation of R-loops is tightly regulated, and cells have developed several mechanisms to resolve them or prevent their formation and/or accumulation. The nucleolar DEAD-box helicase DDX21 is of particular interest, because it localises in the nucleolus, where it directly binds rRNAs and small nucleolar RNAs (snoRNAs), promoting rRNA transcription, processing, and modification (Calo et al, 2015). Depletion of DDX21 leads to R-loop accumulation, stalling of RNA polymerases and increased γH2AX foci (Song et al, 2017). Inhibition of RNA polymerase I– or II–mediated transcription leads to disengagement of DDX21 from its target genes (Song et al, 2017). Furthermore, DDX21 has been associated with nucleolar dysfunction, rDNA damage and craniofacial malformations (Calo et al, 2018). It is worth highlighting that abnormal craniofacial growth patterns have also been reported in SMA patients (Houston et al, 1994). For all these reasons, we aimed to study a potential role for DDX21 in SMA.

We began by immunolabeling SMN Δ7 E13 embryonic spinal motor neurons with an anti-DDX21 antibody, and the total nuclear fluorescence intensity along with foci and nucleoplasmic fluorescence intensity were measured. DDX21 is predominantly located in the nucleolus (Xing et al, 2017); therefore, we defined the foci staining as nucleolar whereas the nuclear staining excluding the foci was defined as nucleoplasmic (Fig 7A). Interestingly, we observed a significant decrease in DDX21 foci intensity in SMN-deficient motor neurons when compared with control cells (Fig 7D), but no difference in DDX21 signal in the nucleus and nucleoplasm (Fig 7B and C). We then used iPSC-derived motor neurons (iPSC-MNs) isolated from two SMA patients and two healthy individuals, to validate these findings in the human context (Fig 7E). Again, SMA iPSC-MNs exhibited reduced intensity of DDX21 foci (Fig 7J and K), and no difference in DDX21 signal in nucleus and nucleoplasm (Fig 7F–I). Importantly, embryonic cortical neurons did not show any significant difference for DDX21 foci intensity between SMA and controls, reinforcing that this is a cell type-specific phenotype (Fig 8A–D). Out of curiosity, we also used fibroblasts derived from an SMA type I patient and a healthy individual. Fibroblasts are cycling cells as opposed to post-mitotic neurons. Therefore, for consistency, cells in their S phase were excluded from the quantification by labelling them with the S phase-specific marker cyclin A2 (Loukil et al, 2015). More specifically, fibroblasts were double-stained with DDX21 and cyclin A2 antibodies and only cyclin A2–negative cells were analysed. Similar to cortical neurons, the intensity of DDX21 foci staining in SMA fibroblasts was unchanged (Fig 8E–H). Taken together, our data suggest that the consequences of rDNA instability are more prominent in SMN-deficient spinal motor neurons.

To investigate the impact of DDX21 deficiency on motor neuron rDNA integrity, we knocked down DDX21 in iPSC-MNs isolated from three healthy individuals. Depletion of DDX21 led to increased nucleolar γH2AX levels in those neurons (Fig 9A-L; γH2AX nucleolar levels are quantified in C,G,K). The nucleolar damage caused by DDX21 knockdown appears to be R-loop mediated (Fig S10). This finding suggests that the motor neuron specific deficiency of DDX21 in SMA, may exacerbate the rDNA damage phenotype seen in SMN-deficient motor neurons and may contribute to their neurodegeneration.

# Discussion

In this study, we highlighted a novel role for SMN in SMA pathology. We showed that SMN-deficient cells exhibit increased R-loop–mediated rDNA damage resulting in nucleolar disruption, particularly in motor neurons. We found that nucleolar disruption was linked to the loss of SMN's function to resolve R-loops. Importantly, we demonstrated that SMN interacts with RNA polymerase I, and lack of SMN had severe repercussions on ribosomal biogenesis, suggesting that defective translation may be a critical component of SMA pathogenesis.

That SMN deficiency leads to transcription-mediated DNA instability was previously described. For instance, Zhao and colleagues showed first that SMN interacts with RNA polymerase II, thereby recruiting SETX to resolve RNA/DNA hybrids at transcription termination sites (Zhao et al, 2016). Loss of either SMN or SETX leads to R-loop accumulation, causing increased DNA damage (Mischo et al, 2011; Jangi et al, 2017; Kannan et al, 2018, 2020, 2022). Studies in yeast have shown that Sen1 protein, the yeast homolog of human senataxin, is involved in transcription termination of the 35S pre-rRNA (Ursic et al, 2004; Kawauchi et al, 2008). In addition to this, mutations of *Sen1* gene in yeast lead to R-loop accumulation over rDNA genes (Chan et al, 2014). Our results using both biochemical and cell-based assays showed that SMN also interacts with RNA polymerase I. In particular, our immunoprecipitations demonstrated that SMN, RNA polymerase I, and SETX are all part of one complex. In other words, SMN would interact with RNA polymerase I and recruit SETX to resolve R-loops originating during rDNA transcription. SMN-deficient cells recruit SETX less efficiently; resulting in R-loop accumulation and increased rDNA damage in the form of DSBs. DSBs are highly toxic lesions that need to be repaired accurately to avoid chromosomal instability, a hallmark of various disorders. Mammalian cells respond to DSBs by initiating a complex DNA damage response (DDR) that involves detection, signalling, and ultimately

---

neurons were stained to reveal overall morphology (beta-III-tubulin, magenta) and nuclear integrity (Hoechst, blue). Protein synthesis was visualized by labelling newly synthesized proteins with L-azidohomoalanine (AHA, green). SMA motor neurons were transduced with Ad-hSETX or lentiviral vector survival motor neuron (SMN) FL before staining. Scale bars represent 10 $\mu m$. **(D)** AHA fluorescence intensity values normalised to control samples. Bar graphs of mean ± SEM #$P$ < 0.05 (comparing SMA cells treated with lentiviral vector-SMN and SMN untreated cells), ##$P$ < 0.01 (comparing SMA cells treated with Ad-SETX and SMA untreated cells) and **$P$ < 0.01 (comparing SMA and control untreated cells). One-way ANOVA analysis followed by Tukey's multiple comparisons test; F (1.615, 3.229) = 13.79. $P$ = 0.0273. The data were collected from N = 3 and were normally distributed.

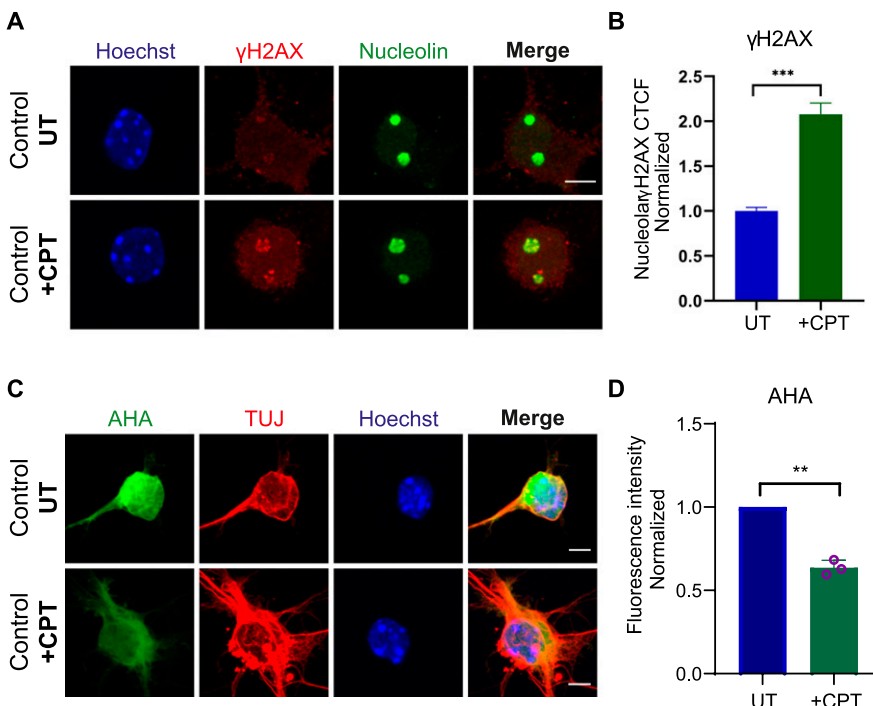

**A**

Hoechst | γH2AX | Nucleolin | Merge

Control UT

Control +CPT

**B**

γH2AX

***

Nucleolar γH2AX CTCF Normalized

2.5
2.0
1.5
1.0
0.5
0.0

UT    +CPT

**C**

AHA | TUJ | Hoechst | Merge

Control UT

Control +CPT

**D**

AHA

**

Fluorescence intensity Normalized

1.5

1.0

0.5

0.0

UT    +CPT

**Figure 6. Extended exposure of control embryonic motor neurons to low levels of camptothecin (CPT) results in increased ribosomal DNA damage and subsequent translation impairment.**
**(A)** Control wild-type embryonic motor neurons were treated with 50 nM CPT for 4 d. CPT-treated and untreated wild type embryonic motor neurons were double-stained with γH2AX (red) and nucleolin (green). Scale bars represent 5 μm. **(B)** Corrected fluorescence intensity of nucleolin-positive γH2AX signal was quantified. Briefly, for each replicate, we calculated the average value of all the UT points and used this as a normalizer. We then plotted all the normalised points from all three replicates in a single graph and performed Mann–Whitney non-parametric test. Bar graphs of mean ± SEM (N = 3). ~150 cells analysed/replicate. ***$P \leq 0.001$. **(C)** Protein synthesis was visualized by labelling newly synthesized proteins with AHA (green). Scale bars represent 10 μm. **(D)** AHA fluorescence intensity values in CPT treated cells normalised to untreated samples. Bar graphs of mean ± SEM (N = 3). **$P < 0.01$; paired two-tailed $t$ test ($P$ = 0.0049). Nuclei counted = 50/replicate.

repair. DSBs in rDNA activate a distinct signalling and chromatin response (Korsholm et al, 2019). When rDNA DSBs fail to reach a threshold level, they are rapidly repaired by non-homologous end joining (NHEJ), involving DNA-dependent protein kinase (DNA-PK) and the ligase XRCC4, whereas RNA polymerase I (Pol-I)-mediated transcription continues (Harding et al, 2015). Interestingly, chronically low levels of SMN result in DNA-PK catalytic subunit (DNA-PKcs) deficiency (Kannan et al, 2018). It is likely that DNA-PKcs deficiency in SMA cells may cause defects in NHEJ-mediated repair, leading to rDNA DSBs accumulation. When the breaks persist, the kinase ataxia-telangiectasia mutated is activated, which silences RNA polymerase I transcription. Transcriptional inhibition initiates nucleolar cap formation at the nucleolar periphery. Next, cap formation facilitates the recruitment of factors required for homologous recombination (HR), which are normally excluded from the nucleolus. Mobilization of rDNA DSBs to the nucleolar periphery is believed to serve as a mechanism to physically separate rDNA repeats from different chromosomes before HR to prevent inter-chromosomal recombination (Mangan et al, 2017). rDNA DSBs are repaired by HR throughout the cell cycle. In G1 phase, HR is templated by other repeats in *cis* (Kruhlak et al, 2007; Harding et al, 2015; van Sluis & McStay, 2015). Therefore, even postmitotic neurons, such as motor neurons use HR for the repair of persistent DSBs.

rDNA DSBs are associated with inhibition of RNA polymerase I–mediated transcription in the proximity to the breaks (Kruhlak et al, 2007). Our results suggest that loss of SMN may induce nucleolar transcriptional silencing, as we detected reduced rRNA levels in SMN-deficient cells. Notably, this reduction was more profound in spinal motor neurons than other neuronal subtypes, such as cortical neurons, suggesting a certain degree of cell selectivity

for this mechanism. Consistently, reduced ribosome numbers and subsequent translational defects have been reported in SMA motor neurons by other groups (Bernabo et al, 2017; Tapia et al, 2017). Aberrant translational repression has emerged as a common feature across multiple neurodegenerative disorders, as reviewed by Lehmkuhl and Zarnescu (2018). However, the mechanisms underlying translational repression are not fully understood. In SMA, it has been reported that SMN depletion leads to an impairment of translation-related transcripts and to consequent defects in ribosomal biology (Bernabo et al, 2017; Lauria et al, 2020). Here, we unravelled a new potential mechanism through which SMN depletion leads to translational defects. We suggest that SMN deficiency leads to R-loop–mediated rDNA damage followed by inhibition of nucleolar transcription. This transcriptional silencing results in low levels of rRNAs, which in turn lead to reduced ribosome pool and a subsequent translational repression. We demonstrate that translational impairment in SMA cells can be rescued by increasing either SMN or SETX protein levels. These findings support the role of SMN in resolving R-loops and safeguarding genome integrity to prevent translational defects in SMA. Our finding is further reinforced by the observation that camptothecin, a drug that transiently stabilizes nucleolar R-loops (Shen et al, 2017), leads to increased rDNA damage and reduced translation in healthy neurons, thus mirroring the pathological mechanism underlying SMA cells.

It remains unclear why defects in the ubiquitously expressed SMN protein lead to disruptions that are largely restricted to spinal motor neurons. We speculate that the translation phenotype may be aggravated in SMN-deficient spinal motor neurons because of their low levels of nucleolar DDX21 protein. Particularly, we hypothesize that reduced DDX21 levels in SMA motor neurons could

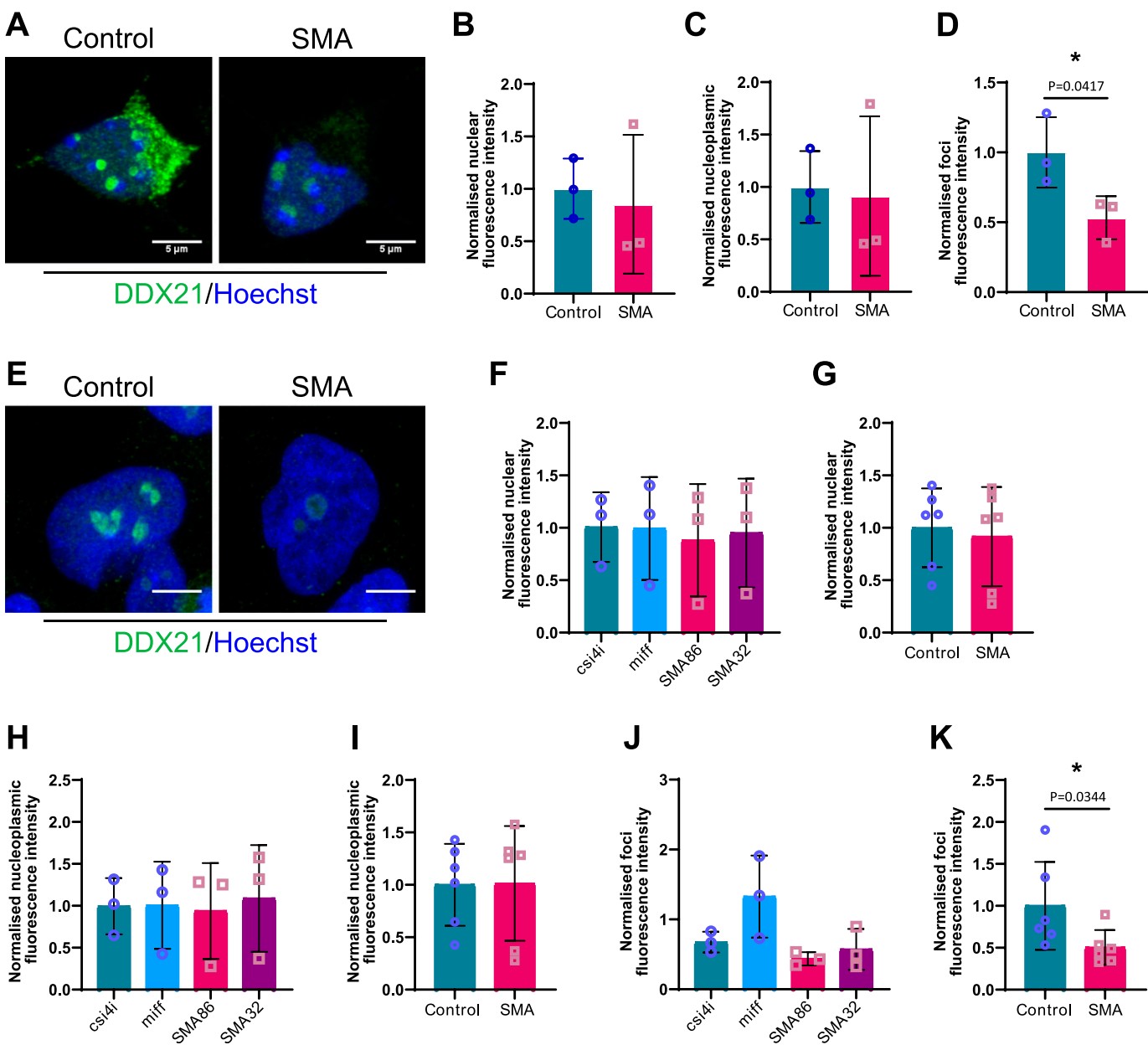

**Figure 7. Survival motor neuron (SMN)–deficient motor neurons exhibit reduced DDX21 levels in a cell autonomous manner.**
**(A)** p75 enriched motor neurons derived from SMNΔ7 and wild type E13 embryos were labelled with DDX21 antibody, at DIV7. Scale bars represent 5 μm. **(B, C, D)** Nuclear, (C) nucleoplasmic and (D) foci fluorescence intensity of DDX21 staining normalised to control. Bar graphs of mean ± s.d. *P < 0.05; paired two-tailed t test (P = 0.0417). **(E)** iPSC-derived motor neurons isolated from healthy individuals (csi4i and miff) and spinal muscular atrophy (SMA) type I patients (SMA86 and SMA32) were labelled with DDX21 antibody. Scale bars represent 5 μm. **(F)** Nuclear fluorescence intensity of DDX21 staining normalised to the average of control samples. **(G)** Nuclear fluorescence intensity where control samples (csi4i and miff) and SMA samples (SMA68 and SMA32) have been pulled together. **(H)** Nucleoplasmic fluorescence intensity of DDX21 staining normalised to the average of control samples. **(I)** Nucleoplasmic fluorescence intensity where control samples (csi4i and miff) and SMA samples (SMA68 and SMA32) have been pulled together. **(J)** Foci fluorescence intensity of DDX21 staining normalised to the average of control samples. **(K)** Foci fluorescence intensity where control samples (csi4i, miff) and SMA samples (SMA68, SMA32) have been pulled together. Bar graphs of mean ± s.d. *P < 0.05; paired two-tailed t test (P = 0.0344).

synergistically contribute, either directly or indirectly, to exacerbate the protein defects detected in SMA. DDX21 is a nucleolar helicase, the depletion of which leads to R-loop accumulation, stalling of RNA polymerases and increase in γH2AX foci (Song et al, 2017). Given that DDX21 facilitates the resolution of R-loops (Song et al, 2017), reduced levels of DDX21 in the nucleolus of SMN-deficient motor

neurons may lead to a further increase in nucleolar R-loops and rDNA damage, thus impacting on protein synthesis. Knockdown of DDX21 in healthy iPSC-MNs led to increased nucleolar γH2AX levels, reinforcing our hypothesis. Alternatively, DDX21 may play a more direct role in ribosome biology by reducing rRNA levels upstream of translation. It has been reported that DDX21 directly binds to rRNA

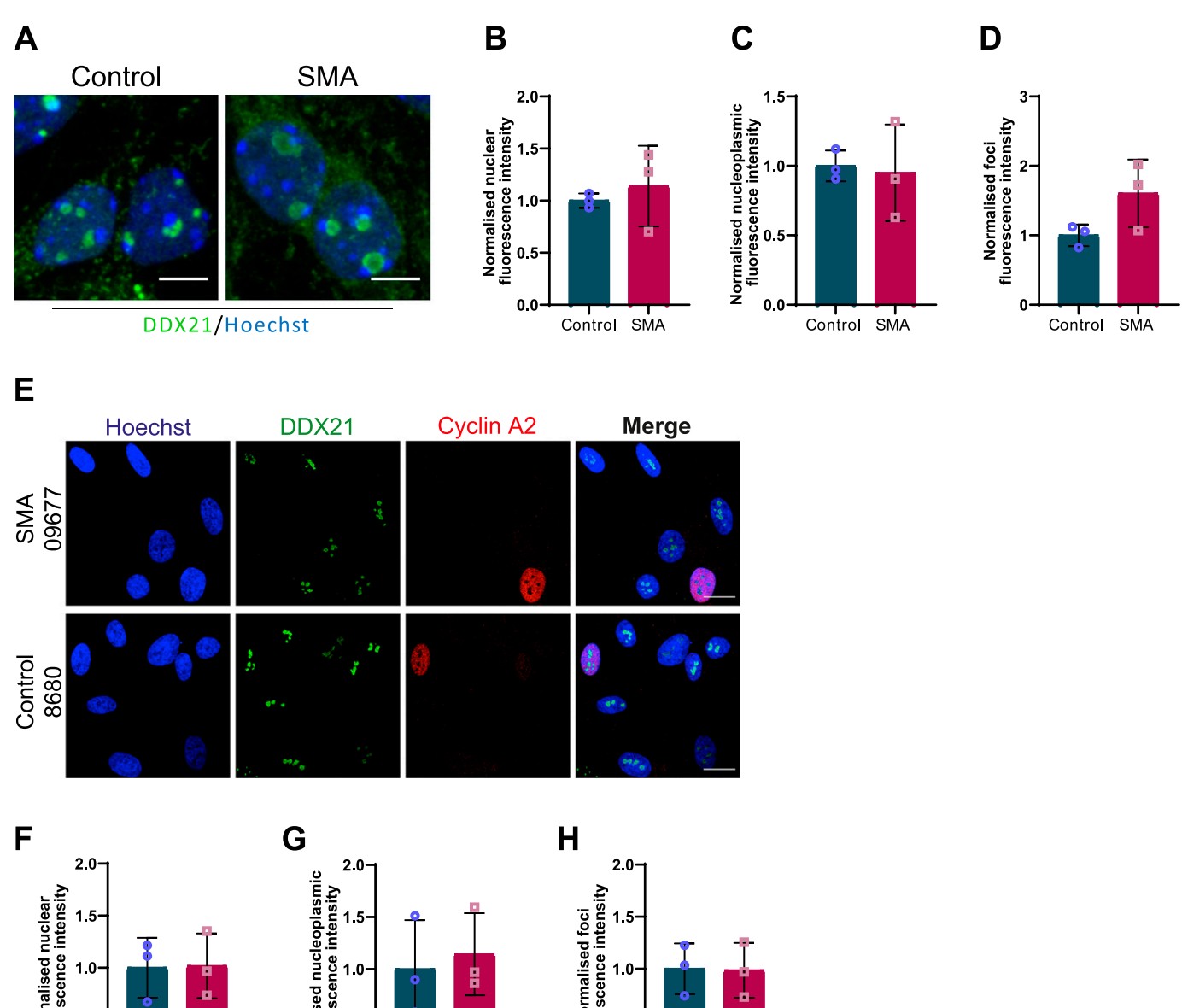

**Figure 8. DDX21 levels in spinal muscular atrophy (SMA) embryonic cortical neurons and fibroblasts derived from SMA patients.**
**(A)** SMA and control embryonic cortical neurons were labelled with DDX21 antibody, at DIV7. Scale bars represent 5 $\mu$m. **(B, C, D)** Nuclear (B), nucleoplasmic (C), and foci (D) fluorescence intensity of DDX21 staining is presented normalised to control. Bar graphs of mean ± s.d (N = 3). ns, not significant (*P* > 0.05). **(E)** Fibroblasts derived from SMA type I patient (SMA) and a healthy individual (control) were doubled-stained with DDX21 and cyclin A1. Scale bars represent 10 $\mu$m. Cyclin A1–positive cells were excluded from the analysis. **(F, G, H)** Nuclear (F), nucleoplasmic (G), and foci (H) fluorescence intensity of DDX21 staining is presented normalised to control. Bar graphs of mean ± s.d (N = 3). ns, not significant (*P* > 0.05).

and snoRNAs, promoting rRNA transcription and processing (Calo et al, 2015). Reduced levels of DDX21 in motor neurons may lead to insufficient binding of DDX21 to rDNA loci resulting in inhibition of rRNA transcription. This possible contribution of DDX21 in SMA pathogenesis requires further investigation to elucidate the exact underlying mechanism.

# Materials and Methods

### Cells and cell culture maintenance

Primary fibroblast cell lines from SMA type I patients (GM03813, GM09677, and GM00232) and age- and gender-matched healthy

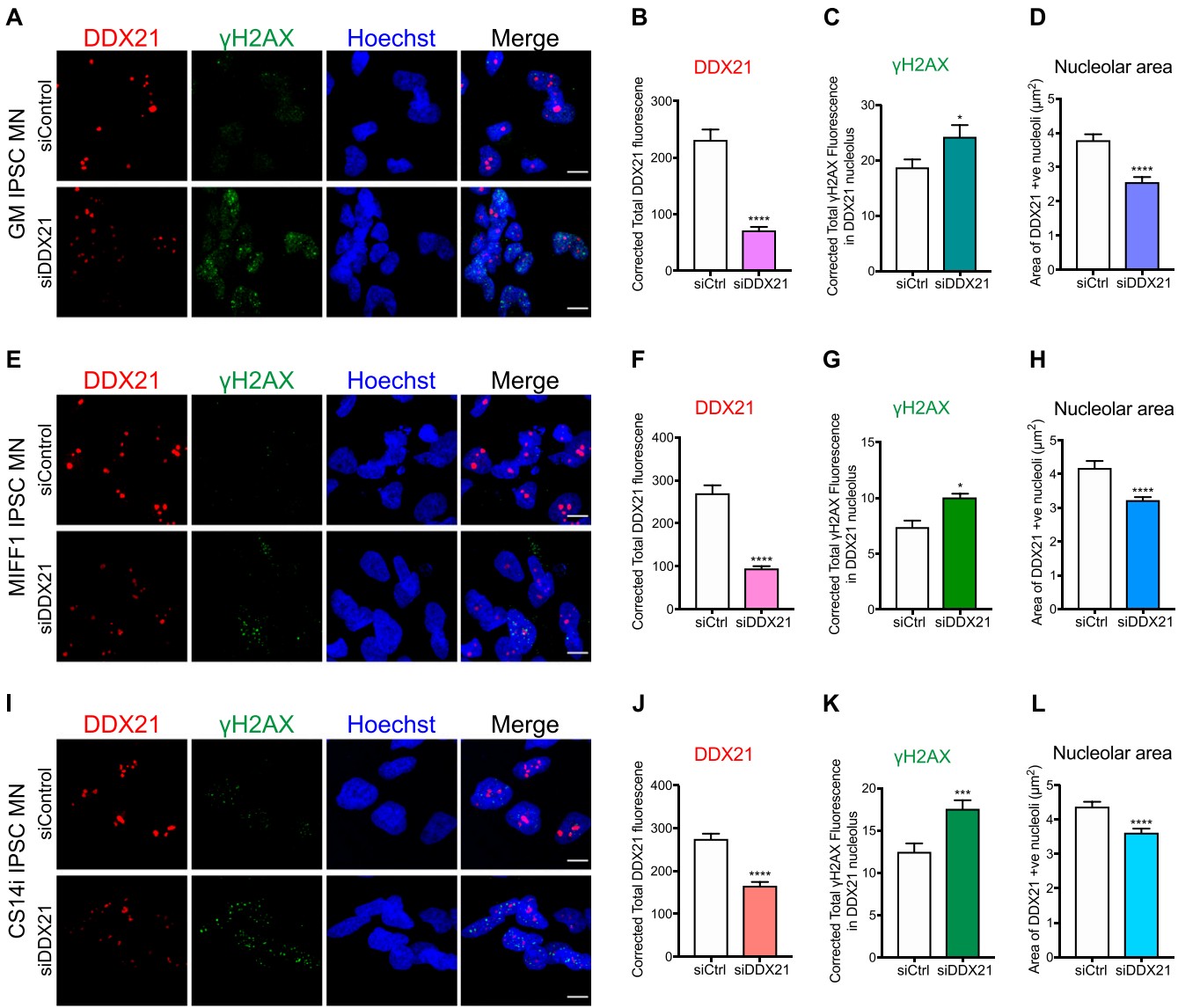

**Figure 9. DDX21 knockdown in iPSC-derived motor neurons reduces nucleolar size and increases nucleolar DNA damage.**
Three independent control iPSC-derived motor neuron lines (GM, MIFF1, and CS14i) were treated with 1 $\mu$M non-targeting control Accell siRNA or 1 $\mu$M DDX21-targeted Accell siRNA according to the manufacturer's instructions. 4 d post-transfection cells were fixed and stained with anti-DDX21 and anti-$\gamma$H2AX antibodies. **(A)** Representative images of GM control IPSC-motor neurons (MNs) treated with control or DDX21 targeted Accell siRNA. **(B)** Corrected fluorescent intensity of DDX21-positive nucleolar structures in GM MNs (mean ± SEM; unpaired $t$ test, ****$P \le 0.0001$; N [nucleoli] = siCtrl 298, siDDX21 145). **(C)** Corrected fluorescent intensity of $\gamma$H2AX signal within DDX21 positive nucleolar structures in GM MNs (mean ± SEM; unpaired $t$ test, *$P \le 0.05$). **(D)** Surface area of DDX21-positive nucleolar structures in GM MNs (mean ± SEM; unpaired $t$ test, ****$P \le 0.0001$). **(E)** Representative images of MIFF1 control IPSC MNs treated with control or DDX21 targeted Accell siRNA. **(F)** Corrected fluorescent intensity of DDX21-positive nucleolar structures in MIFF1 MNs (mean ± SEM; unpaired $t$ test, ****$P \le 0.0001$; N (nucleoli) = siCtrl 160, siDDX21 490). **(G)** Corrected fluorescent intensity of $\gamma$H2AX signal within DDX21-positive nucleolar structures in MIFF1 MNs (mean ± SEM; unpaired $t$ test, *$P \le 0.05$). **(H)** Surface area of DDX21 positive nucleolar structures in MIFF1 MNs (mean ± SEM; unpaired $t$ test, ****$P \le 0.0001$). **(I)** Representative images of CS14i control iPSC MNs treated with control or DDX21 targeted Accell siRNA. **(J)** Corrected fluorescent intensity of DDX21-positive nucleolar structures in CS14i MNs (mean ± SEM; unpaired $t$ test, ****$P \le 0.0001$; N (nucleoli) = siCtrl 415, siDDX21 387). **(K)** Corrected fluorescent intensity of $\gamma$H2AX signal within DDX21-positive nucleolar structures in CS14i MNs (mean ± SEM; unpaired $t$ test, ***$P \le 0.001$). **(L)** Surface area of DDX21-positive nucleolar structures in CS14i MNs (mean ± SEM; unpaired $t$ test, ****$P \le 0.0001$). Scale bars = 10 $\mu$m.

individuals (GM00498, GM05658, and GM08680) were obtained from Coriell Cell Repositories. They were grown in DMEM medium containing 10% (vol/vol) fetal bovine serum, 50 $\mu$g/ml uridine, and 1% penicillin/streptomycin. For primary cortical neuron culture, E16 embryos were dissected from carrier pregnant females (SMN$\Delta$7+/+; SMN2; Smn+/−) (Le et al, 2005). DNA was isolated from embryos for genotyping. Cortical neurons were plated on poly-D-lysine coated

plates and maintained in Neurobasal medium supplemented with 2% B27, 0.5 mM GlutaMax, and 1% penicillin/streptomycin. For primary spinal motor neuron cultures, E13 embryos were dissected from carrier pregnant females (SMN$\Delta$7+/+; SMN2; Smn+/−) and the genotype of the embryos established. Cultures of embryonic lower motor neurons were prepared as described in Ning et al (2010) and Wiese et al (2010). Briefly, spinal motor neurons were isolated using

the p75 immunopanning method. Cells were plated on poly-D-ornithine and laminin coverslips and maintained in Neurobasal medium supplemented with 2% B27, 2% horse serum, 0.5 mM GlutaMax, 25 μM 2-mercaptoethanol, 5 ng/ml CNTF, 1 ng/ml GDNF, 5 ng/ml BDNF, and 1% penicillin/streptomycin. Finally, SMA and control iPSC cells were differentiated into motor neurons and cultured according to a published protocol (Du et al, 2015).

### Electrophysiology

Differentiated iPSC-derived motor neurons were plated onto 13 mm coverslips at a density of 10,000 cells per coverslip. Electrophysiology was carried out at 6 wk of differentiation. All recordings were performed at room temperature and all reagents for solutions were purchased from Sigma-Aldrich. Electrodes for patch clamping were pulled on a Sutter P-97 horizontal puller (Sutter Instrument Company) from borosilicate glass capillaries (World Precision Instruments). Coverslips were placed into a bath on an upright microscope (Olympus) containing the extracellular solution at pH7.4 composing of 150 mM NaCl, 5.4 mM KCl, 2 mM $MgCl_2$, 2 mM $CaCl_2$, 10 mM Hepes, and 10 mM glucose osmolarity, ~305 mOsm/Kg. Whole-cell current clamp recordings were performed using an Axon Multi-Clamp 700B amplifier (Axon Instruments) using unpolished borosilicate pipettes placed at the cell soma. Pipettes had a resistance of 4–6 MΩ when filled with intracellular solution of 140 mM $K^+$-gluconate, 10 mM KCl, 1 mM $MgCl_2$, 0.2 mM EGTA, 9 mM NaCl, 10 mM Hepes, 0.3 mM Na+-GTP, and 3 mM Na-ATP adjusted to 298 mOsm/Kg at pH7.4. For both solutions glucose, EGTA, Na+-GTP, and Na+-ATP were added fresh on each day of the experiment. To identify the motor neurons present, cells were visualized using the microscope 40× objective, and those with a triangular cell body and processes to indicate a motor neuronal morphology were selected. To measure depolarized evoked action potential firing in the cells using a five-step protocol for a duration of 500 ms, injecting current from −40 pA, every 20 pA. Recordings were acquired at ≥10 kHz using a Digidata 1440A analogue-to-digital board and pClamp10 software (Axon Instruments). Electrophysiological data were analysed using Clampfit10 software (Axon Instruments). Firing magnitude 20 mV and higher were included for analysis.

### Viral vectors and transduction

Codon optimised SMN (coSMN) (generated by Geneart AG) was subcloned into a self-inactivating lentiviral (SIN-W-PGK) vector using standard cloning methods (Valori et al, 2010). Lentiviruses were propagated in HEK293T cells using the calcium phosphate method (Déglon et al, 2000). Viral titres were measured by qPCR. Genomic DNA isolated from transduced HeLa cells was used as a template for qPCR with Woodchuck Hepatitis Virus Posttranscriptional Regulatory Element (WPRE) primers to assess the number of copies of stably integrated lentiviruses. An LV carrying GFP of known biological titre (FACS titration) was used as a reference. Primary motor neurons were transduced with lentiviral vector coSMN MOI of 3. A high-titered purified recombinant adenovirus (Adenovirus-type 5 dE1/E3) carrying human *SETX* gene (Ad-h-SETX) was generated by Vector Biolabs. Primary motor neurons were transduced at DIV 2 with a MOI of 28.

### Accell siRNA knockdown

Dharmacon Accell siRNA was purchased from Horizon Discovery. IPSC motor neurons were treated with non-targeting control Accell siRNA or DDX21-targeted Accell SMARTPool siRNA according to the manufacturer's instructions. Briefly, reconstituted Accell SMART-Pool siRNAs were diluted in IPSC motor neuron culture media, and added directly to the cells at a final concentration of 1 μM. Cells were treated with siRNA for 4 d, at which time they were fixed in 4% PFA and processed for immunocytochemistry. Cells were given 50% media changes every 2 d during motor neuron differentiation and siRNA treatment.

The DDX21-targeted Accell SMARTPool siRNA sequences were as follows:

A-011919-13: GAAUUAAGUUCAAACGAAU
A-011919-14: CCAUGAUCUUGCAGUGCUC
A-011919-15: GGAAGAAUAUCAGUUAGUA
A-011919-16: GUAGUAUGUUUGAUGUGGA.

### Immunocytochemistry

Cells cultured on glass coverslips were fixed in 4% PFA (or 1:1 methanol: acetone for R-loop staining) for 10 min, washed with PBS and permeabilized with 0.5% Triton X-100 in PBS. Cells were then blocked with 3% BSA in PBS for 30 min at room temperature and incubated with primary antibodies diluted in blocking buffer for 2 h at room temperature. The primary antibodies used in this study included SMN (1:1,000, 610646; BD), γH2AX (1:1,000, 05-636; Merk Millipore), DNA/RNA hybrids [S9.6] (Boguslawski et al, 1986) (1:2,000; Kerafast), Nucleolin (1:2,000, ab22758; Abcam), DDX21 (1:500, NB100-1718; Novus Biologicas). Finally, the cells were counterstained with secondary Alexa-Fluor conjugated antibodies (1:1,000; Invitrogen) diluted in a blocking buffer and incubated for 1 h at room temperature. Hoechst stain was used to visualize nuclei. Images were captured using a Leica SP5 confocal microscope, primarily with a 63× 1.4 NA oil immersion objective.

### Immunohistochemistry

Mice were terminally anesthetized and transcardially perfused as previously described (Ning et al, 2010). Spinal cords were then cryoprotected in 30% sucrose, embedded in Optimal Cutting Temperature matrix and 16-μm lumbar spinal cord sections were cut on a sliding cryostat microtome (Leica). For R-loops staining, S9.6 (1:1,000; Kerafast) antibody was used. Before primary antibody incubation, antigen retrieval was performed in 10 mM Tris for 30 min in a pressure cooker. Visualisation of the primary antibodies was enabled by use of the intelliPATH FLX Detection Kit, according to the manufacturer's protocol. Human postmortem spinal cords were also stained for R-loops in a similar way.

### Western blotting

Unless otherwise indicated, total cellular protein was extracted as follows: cells were rinsed in ice-cold PBS and lysed with a nuclear and cytoplasmic lysis buffer (5% Tris–HCl [pH 7.4], 1% NP-40, 0.5%

Sodium deoxycholate, 0.01% SDS, 150 mM NaCl, and 0.2 mM EDTA) supplemented with 1% Protease Inhibitor Cocktail (Sigma-Aldrich). Protein concentration was determined by BCA assay (Thermo Fisher Scientific). Protein equivalents from each sample were subjected to SDS–PAGE followed by immunoblotting. 5% milk in TBST was used for blocking and incubation with antibodies.

## Dot-blot analysis of R-loops

Genomic DNA was extracted from HeLa cells using the GenElute Mammalian Genomic DNA Miniprep Kit (Sigma-Aldrich). Genomic DNA (1 μg) was spotted to Amersham Hybond-N+ positively charged nylon membrane presoaked in PBS using the 96-well Bio-Dot Microfiltration apparatus (Bio-Rad). DNA was UV-crosslinked to the membrane with 120 mJ/cm$^2$ using a UV TL-2000 Translinker. Membranes were blocked for 1 h in PBS with 5% fat-free milk (Sigma-Aldrich) and 0.1% Tween-20 (PBS-T) before incubation with mouse S9.6 (anti-R-loop) antibody (1 μg/ml; 1:1,000 dilution) in blocking buffer overnight at 4°C. Membranes were washed three times for 10 min in PBS-T before incubation with goat anti-mouse HRP secondary antibodies (1:5,000 dilution) in PBS-T for 1 h at room temperature. After 3 × 10 min washes in PBS-T membranes were prepared for chemiluminescent signal detection with SuperSignal West Pico Chemiluminescent substrate kit (Thermo Fisher Scientific) according to the manufacturer's instructions. Signals were detected and densitometric quantification of dots were performed on a LI-COR Odyssey Fc Imager (LI-COR).

## γH2AX-ChIP

Pelleted SMA type I and healthy fibroblasts (roughly 5 × 10$^6$ cells/group) were resuspended in 10 ml PBS and chemically crosslinked by the addition of 270 μl 37% PFA while rotated for 10 min on ice. The cross-link reaction was then stopped by the addition of 1 ml 1.25 M glycine and a further incubation at 4°C for 5 min. The crosslinked cells were pelleted by centrifugation at 500g for 5 min at 4°C, washed once with PBS and re-suspended in 500 μl ChIP lysis buffer (50 mM HEPES-KOH, pH 7.5, 140 mM NaCl, 1 mM EDTA pH 8, 1% Triton X-100, 0.1% sodium deoxycholate, and 0.1% SDS). The cells were lysed for 30 min while shaking in a thermomixer set at 4°C. The lysed cells were then sonicated at 4°C for 35 cycles (30 s on, 30 s off) using a Diagenode's Bioruptor to solubilize and shear crosslinked DNA. After the sonication the samples were centrifuged at 4°C for 10 min at maximum speed and the supernatant was collected. 10% of each sample was kept as input. The remaining 90% was then subjected to immunoprecipitation (IP). Briefly, 30 μl of protein G magnetic beads (Invitrogen) were washed two times with PBS/0.02% Tween-20 and incubated for 1 h at 4°C with 5 μg of γH2AX (Merk Millipore) or mouse IgG antibody. Beads were washed two times with RIPA buffer. Meanwhile, 400 μl of the cell lysates were diluted 1:1 with 400 μl of RIPA lysis buffer and then loaded onto the beads. 2-h incubation at 4°C was followed. At the end of the incubation, beads were washed four times with RIPA buffer and once with elution buffer (1% SDS, 0.1 M NaHCO₃). Bound complexes were eluted from the beads by heating at 65°C in a thermomixer and crosslinking was reversed by overnight incubation at the same temperature. Immunoprecipitated DNA and 10% input samples

**Table 1. Primers used for γH2AX ChIP-qPCR.**

| Primer | Sequence |
|---|---|
| Human RPL32 F | GAAGTTCCTGGTCCACAACG |
| Human RPL32 R | GCGATCTCGGCACAGTAAG |
| Human 18 S F | ATGGCCGTTCTTAGTTGGTG |
| Human 18S R | CGCTGAGCCAGTCAGTGTAG |
| Human 5.8S F | GACTCTTAGCGGTGGATCACTC |
| Human 5.8S R | GACGCTCAGACAGGCGTAG |
| Human 28S F | CAGGGGAATCCGACTGTTTA |
| Human 28S R | ATGACGAGGCATTTGGCTAC |

were then purified by treatment with RNase A, proteinase K and phenol/chloroform extraction. The DNA samples were then subjected to qPCR. The qPCR primer sequences are shown in Table 1.

## HB-GFP ChIP-qPCR

Primary fibroblasts from SMA patient (GMO9677) and healthy individual (GMO8680) were transduced with a lentivirus carrying HB-GFP sequence at a MOI of 30; an empty vector-GFP virus was used as an extra control. Cells were maintained in DMEM media and 6 T175 flasks were used from each cell line for the experiment. Cells were crosslinked at room temperature for 10 min with PFA (1% final concentration); reaction was quenched with Glycine (125 mM final concentration) for 5 min. Cells were harvested by scrapping and pellets lysed in 500 μl of ChIP lysis buffer (50 mM HEPES-KOH, pH 7.5, 140 mM NaCl, 1 mM EDTA pH8, 1% Triton X-100, 0.1% sodium deoxycholate, 0.1% SDS, and 1X protease inhibitors) for 10 min at 4°C (on ice). Next, lysates were sonicated for 35 cycles (30 s ON-30 s OFF) at 4°C on the Diagenode Bioruptor Pico, centrifuged at 8,000g and 4°C for 10 min and supernatants were recovered. A small volume (50 μl) was used to determine DNA concentration and fragment size: crosslinks were reversed and DNA was purified using phenol–chloroform, fragment sizes around 200 bp were confirmed on 1% agarose gels. The rest of the lysate was used for the immunoprecipitation (~40 μg of DNA per sample), samples were diluted 1:10 in RIPA buffer (50 mM Tris–HCl, pH 8, 150 mM NaCl, 2 mM EDTA pH 8, 1% NP-40, 0.5% sodium deoxycholate, 0.1% SDS, and 1X protease inhibitors) and 30 μl GFP-Trap beads (ChromoTek) were added. After overnight incubation at 4°C, the beads were washed once with low salt wash buffer (0.1% SDS, 1% Triton X-100, 2 mM EDTA, 20 mM Tris–HCl, pH 8.0, and 150 mM NaCl); high salt wash buffer (0.1% SDS, 1% Triton X-100, 2 mM EDTA, 20 mM Tris–HCl, pH 8.0, and 300 mM NaCl); and LiCl wash buffer (0.25 M LiCl, 1% NP-40, 1% sodium deoxycholate, 1 mM EDTA, and 10 mM Tris–HCl, pH 8.0). R-loops were eluted at 30°C and 800g for 30 min in Elution buffer (1% SDS and 100 mM NaHCO₃). After the IP, cross-links were reversed from eluates and inputs; DNA was purified by phenol–chloroform extraction, and quantified. The qPCR was prepared using SensiMix SYBR No-ROX kit, and carried out in the QIAGEN Rotor Gene. The primers used for these reactions are listed in Table 2.

**Table 2. Primers used for HB-GFP ChIP-qPCR.**

| Primer | Sequence |
|---|---|
| Actin 5' pause (F) | TTA CCC AGA GTG CAG GTG TG |
| Actin 5' pause (R) | CCC CAA TAA GCA GGA ACA GA |
| R#7 (F) | GAC ACT TCG AAC GCA CTT G |
| R#7 (R) | CTC AGA CAG GCG TAG CCC CG |
| Human 5.8S F | GACTCTTAGCGGTGGATCACTC |
| Human 5.8S R | GACGCTCAGACAGGCGTAG |
| Human 28S F | CAGGGGAATCCGACTGTTTA |
| Human 28S R | ATGACGAGGCATTTGGCTAC |

## In situ PLA

In situ PLA was performed as recommended by the manufacturer (DuolinkII kit; Olink Bioscience AB). Briefly, HEK293T cells were fixed and permeabilized. Primary antibodies were diluted (SMN 1:1,000, RNA polymerase I [Abcam] 1:200) in antibody diluent and incubated for 1 h at room temperature. The negative control consisted of using only one primary antibody (RNA polymerase I). The cells were washed twice for 2 min in Tris-buffered saline with 0.05% Tween. The PLA probes (Rabbit-MINUS and Goat-PLUS; Olink BioScience AB) were incubated for 90 min at 37°C. Subsequent steps were performed using the detection reagent Orange (Sigma-Aldrich) according to the DuolinkII kit protocol. In situ PLA signals were visible as dots with the RITC filter on the microscope. Nuclei were counterstained with DAPI to be able to select image position.

## pEGFP-SMN constructs

pEGFP-SMN FL and pEGFP-SMN Δ7 plasmids were a generous gift from Dr. Suzie Lefebvre (Université Paris Descartes) (Renvoise et al, 2006). pEGFP-SMN E134K was a generous gift from P. Lomonte (Université Claude Bernard Lyon 1) (Sabra et al, 2013). pEGFP-SMN Δ3 was generated by an inverse PCR deletion method (Wang & Wilkinson, 2001) using pEGFP-SMN FL as a template.

## qRT-PCR

Cells were harvested and RNA was extracted using RNeasy Mini kit (QIAGEN) or Direct-zol RNA Miniprep (Zymo Research) according to the manufacturer's guidelines. The concentration of extracted RNA was then measured using NanoDrop 1000 and the RNA was then subjected to quantitative reverse transcription PCR (qRT-PCR) using QuantiFast SYBR Green RT-PCR kit (QIAGEN). 10 ng of RNA (or 1 ng of RNA for rRNA analysis) was added to 5 $\mu$l 2× QuantiFast SYBR Green RT-PCR Master Mix along with the appropriate forward and reverse primers (Table 3) (the final concentration of each primer was 1 $\mu$M), 1 $\mu$l QuantiFast RT Mix and H$_2$O to make a final volume of 10 $\mu$l. qRT-PCR was performed by CFX96 Real-Time System C1000 Touch Thermal Cycler (Bio-Rad) using the program below:

  i. 50°C 10 min
  ii. 95°C 5 min

**Table 3. Primers used for qRT-PCR.**

| Primer | Sequence |
|---|---|
| mouse 45S F | CTCTTCCCGGTCTTTCTTCC |
| mouse 45S R | TGATACGGGCAGACACAGAA |
| mouse 18S F | CGCGGTTCTATTTTGTTGGT |
| mouse 18S R | AGTCGGCATCGTTTATGGTC |
| mouse 5.8S F | TCGTGCGTCGATGAAGAA |
| mouse 5.8S R | CGCTCAGACAGGCGTAGC |
| mouse 28S F | CCCGACGTACGCAGTTTTAT |
| mouse 28S R | CCTTTTCTGGGGTCTGATGA |
| mouse GAPDH F (reference gene) | CAACTTTGGTATCGTGGAAGGAC |
| mouse GAPDH R (reference gene) | ACAGTCTTCTGGGTGGCAGTG |

  iii. 95°C 10 s
  iv. 60°C 30 s

# Supplementary Information

# Acknowledgements

SF El-Khamisy and M Azzouz made equal contribution through the Wellcome Trust Investigator Award (103844) to SF El-Khamisy and the European Research Council grant (ERC Advanced Award no. 294745) to M Azzouz. M Azzouz is supported by the European Research Council (ERC Advanced Award 294745), MRC Award (MR/G1001492), CureAP4, LifeArc, JPND (MR/V000470/1), ARUK, and IMI Award (945473). SF El-Khamisy was supported by the Wellcome Trust Investigator Award (103844), the Lister Institute of Preventative Medicine Fellowship, and the European Union British Council award (171964603). E Karyka was sponsored by Eve Davis Studentship and CureAP4. The work of L Marrone is supported by a small grant from the British Neuropathological Society. Tissue samples were obtained from Plymouth NHS Trust as part of BRAIN UK, which is funded by the Medical Research Council and Brain Tumour Research. We also thank the Wolfson Foundation for their support in funding the Leica Confocal microscope at SiTraN. The authors would also like to thank Dr Jouni Takalo and Dr Stephen Ebbens (The University of Sheffield) for their guidance in statistical analysis.

## Author Contributions

E Karyka: conceptualization, data curation, formal analysis, validation, methodology, and writing—original draft, review, and editing.
N Berrueta Ramirez: formal analysis and methodology.
CP Webster: conceptualization, formal analysis, and methodology.
PM Marchi: data curation, formal analysis, and methodology.
EJ Graves: formal analysis and methodology.
VK Godena: methodology.
L Marrone: methodology.
A Bhargava: methodology.
S Ray: methodology.
K Ning: methodology.

H Crane: data curation, formal analysis, and methodology.

GM Hautbergue: supervision and methodology.

SF El-khamisy: conceptualization, resources, formal analysis, supervision, funding acquisition, project administration, and writing—review and editing.

M Azzouz: conceptualization, resources, data curation, formal analysis, supervision, funding acquisition, project administration, and writing—review and editing.

**Conflict of Interest Statement**

The authors declare that they have no conflict of interest.

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
