## [Reviewer comments · Life Science Alliance]

Life Science Alliance

SMN-deficient cells exhibit increased ribosomal DNA damage

Evangelia Karyka, Nelly Berrueta Ramirez, Christopher Webster, Paolo Marchi, Emily Graves, Vinay Godena, Lara Marrone, Anushka Bhargava, Swagat Ray, Ke Ning, Hannah Crane, Guillaume Hautbergue, Sherif El-khamisy, and Mimoun Azzouz
DOI: <https://doi.org/10.26508/lsa.202101145>

Corresponding author(s): Mimoun Azzouz, University of Sheffield and Sherif El-khamisy, University of Sheffield

Review Timeline:

Submission Date:	2021-06-28
Editorial Decision:	2021-08-12
Revision Received:	2022-01-10
Editorial Decision:	2022-02-16
Revision Received:	2022-03-31
Editorial Decision:	2022-03-31
Revision Received:	2022-04-04
Accepted:	2022-04-04

Scientific Editor: Novella Guidi

Transaction Report:

August 12, 2021

Re: Life Science Alliance manuscript #LSA-2021-01145-T

Prof. Mimoun Azzouz
University of Sheffield
University of Sheffield
Neurology Unit
Sheffield
United Kingdom

Dear Dr. Azzouz,

Thank you for submitting your manuscript entitled "SMN-deficient cells exhibit increased ribosomal DNA damage" to Life Science Alliance. The manuscript was assessed by expert reviewers, whose comments are appended to this letter. As you will note from the reviewers' comments below, both reviewers are quite positive and think that this work identifies a novel role for SMN in the homeostasis of ribosomal DNA and that is of interest for the SMA field. Reviewer 1 main concerns are about quantification and proper controls to be included. Reviewer 3 main concern is more about the role of decreased DDX21 in motor neurons in contributing to R-loop accumulation. Although this reviewer suggests to perform knockdown and rescue overexpressing experiments to gain more insights into the molecular mechanism, we do not think that the rescue experiment is absolutely necessary for the scope of this manuscript. We encourage though, if doable for the authors, to run the knockdown experiment to evaluate the effect of DDX21 lack in a WT setting, as this would strengthen the existing data. All the other concerns raised by the reviewers should be addressed as well. We, thus, encourage you to submit a revised version of the manuscript back to LSA that responds to all the reviewers' points.

Thank you for this interesting contribution to Life Science Alliance. We are looking forward to receiving your revised manuscript.

Sincerely,

-- Summary blurb (enter in submission system): A short text summarizing in a single sentence the study (max. 200 characters including spaces). This text is used in conjunction with the titles of papers, hence should be informative and complementary to

the title and running title. It should describe the context and significance of the findings for a general readership; it should be written in the present tense and refer to the work in the third person. Author names should not be mentioned.

B. MANUSCRIPT ORGANIZATION AND FORMATTING:

Reviewer #1 (Comments to the Authors (Required)):

Reduced levels of functional survival motor neuron (SMN) protein cause the childhood motor neuron disease spinal muscular atrophy (SMA). Being widely and constitutively active, it remains a mystery as to why low SMN availability primarily impacts the lower motor neurons, leading to muscle wasting and atrophy. The SMN protein has a canonical function in snRNP biogenesis and pre-mRNA splicing; however, several additional roles for the SMN protein have recently been discovered, indicating that SMN is multi-potent.

In the manuscript under review, Karyka and colleagues identify a novel role for SMN in the homeostasis of ribosomal DNA. Several SMA models are first shown to display increased frequency of R-loop (DNA/RNA hybrids) formation, which can lead to deleterious double-stranded DNA breaks. This is shown to be associated with nucleolar defects, increased rDNA damage and perturbed rRNA synthesis and translation. SMN is then shown to interact with RNA polymerase I, which transcribes rDNA, highlighting a possible mechanism for the involvement of rDNA damage in SMA.

Overall, the manuscript is well written, the sum of the data are convincing, and this work is of interest to the field. However, before considering the manuscript for acceptance, I have several points that need addressing:

- Major
- While the phenotype of R-loop accumulation is found across SMA models in Figure 1, not all of them have been convincingly quantified. For example, A) R-loop formation in the mouse spinal cord, B) S9.6 nuclear staining in fibroblasts, and C) nucleolin in fibroblasts and motor neurons. The importance/relevance of these phenotypes cannot be determined without quantification. The same goes for the images in Figures E4 and E5.
 - The PLA images presented in Figure 4C also suffer from a lack of appropriate quantification and controls.
 - The input control is missing from Figure 4E.
 - An empty virus control is missing from the experiments presented in Figure 5C-D.
 - The data presented in Figure 6B are variable and would benefit from an additional experiment or two to confirm the reported phenotype.

- Minor
- SMN protein levels in all of the models presented in Figure 1 should really be shown, like they have for the iPSCs.
 - Red/green immunofluorescent colour-combinations should be avoided (e.g. Fig. 2) due to issues for certain forms of colour-blindness. Please see the following link and amend the figures accordingly: <https://www.ascb.org/science-news/how-to-make-scientific-figures-accessible-to-readers-with-color-blindness/>
 - Please check citations of the EV figures within the text; e.g., EV5 on page 8, should be EV6.
 - Unlike the other transfected SMN proteins, SMN E134K looks to be excluded from the nucleus (Figure EV6) - could this explain the reduced interaction with RNA Pol?
 - The labelling/statistics on Figure 5D are unclear.
 - Please provide all antibody catalogue numbers in the methods.
 - Please include the age of SMN delta 7 mice used in the main text and Figure legends.
 - Please check all scale bars - e.g., Figure 1C: both cannot be 10 µm.
 - Please provide details of any reference genes and primer sequences used in the qPCR experiments.

Reviewer #3 (Comments to the Authors (Required)):

The manuscript by Karyka et al. entitled "SMN-deficient cells exhibit increased ribosomal DNA damage" is very well written and presents new findings that may be interest to scientific community working in the field of spinal muscular atrophy (SMA) and

beyond. Authors show that the SMN interacts with RNA Pol I and ribosomal DNA (rDNA) damage caused by R-loop accumulation in the nucleolus due to SMN deficiency may contribute to SMA pathogenesis. Further, authors show downregulation of DDX21, a DEAD-box containing nucleolar helicase, may contribute to post-mitotic motor neuron degeneration differentiated from iPSCs derived from SMA patients but may not affect cortical neurons or dividing cells derived from SMA patients. Together, these data are very interesting. However, there are a couple of weaknesses that should be eliminated by performing the following experiments to support conclusions and strengthen the manuscript.

1. Authors show that DDX21 is selectively downregulated in motor neurons and its deficiency may contribute to R-loop accumulation, DNA damage and neurodegeneration associated with SMA. (a) Authors should examine the effect of siRNA-based knockdown of DDX21 in iPSC-derived motor neuron from controls. (b) The effect of complementation using overexpression of recombinant DDX21 (adeno or lenti-viral vectors) in SMA motor neurons on the rescue to R-loops and degenerative phenotype.

2. There are inconsistencies reported in the literature about the use S9.6 antibody in immunofluorescence-based quantification of R-loop because S9.6 antibody interacts with dsRNA. I recommend using additional method for R-loop quantitation, a dot blot method using purified genomic DNA to strengthen the finding (Ramirez, P. et al, J. Vis. Exp., 2021; Grunseich, C. et al., Mol. Cell, 2018).

Minor:

Discussion section, last paragraph. Please expand and clarify the statement "Alternatively, DDX21 may play a more direct role in ribosome biology by reducing rRNA levels upstream of translation."

Response to Reviewers' comments**Response to Reviewer 1**

• While the phenotype of R-loop accumulation is found across SMA models in Figure 1, not all of them have been convincingly quantified. For example, A) R-loop formation in the mouse spinal cord, B) S9.6 nuclear staining in fibroblasts, and C) nucleolin in fibroblasts and motor neurons. The importance/relevance of these phenotypes cannot be determined without quantification. The same goes for the images in Figures E4 and E5.

We agree with the reviewer's comment in relation to the missing quantification for some data presented in Figure 1. We therefore revised this figure by adding quantification of the phenotypes as requested under B and C in the reviewer's comment. However, R-loop accumulation in the SMA mouse spinal cord has already been reported by others (Jangi, Fleet et al. 2017, Kannan, Jiang et al. 2020), in particular, the study reported by Kannan et al. (Kannan, Jiang et al. 2020). We therefore moved the images of the mouse spinal cord sections (previous Figure 1C) to a supplementary figure (EV2) in order to prioritise adding quantification to all other panels of Figure 1. Regarding figures EV4 and EV5, we are not really drawing any major conclusions from them, for this reason we haven't included any quantification. In Figure EV4, we simply present the specificity of the R-loop (S9.6) antibody. In our group, we have established an efficient and effective fluorescent immunocytochemistry protocol for characterising R-loops accumulation using S9.6 antibody and we have utilised it in a number of publications (Walker et al. 2017, Jurga et al. 2021), where treatment with RNase H enzyme has also been included. As for Figure EV5, it is well-known that R-loops are considerably enriched in rDNA arrays (Massé, Phoenix et al. 1997, El Hage, French et al. 2010, Nadel, Athanasiadou et al. 2015), and bright S9.6 staining has been reported in mammalian nucleoli before (Ginno, Lott et al. 2012, Sollier, Stork et al. 2014), therefore we do not report any novel data in EV5.

• The PLA images presented in Figure 4C also suffer from a lack of appropriate quantification and controls.

Direct interaction between SMN and RNA polymerase I was tested in HEK293T cells using the in situ proximity ligation assay (PLA) and it was presented in Figure 4C similarly to other published studies (e.g. Renvoise, Querol et al. 2012). We have addressed this request in our revised manuscript by including a graph illustrating the number of puncta per nucleus for each of our conditions (Figure 4C and pages 8 and 35 of the revised manuscript).

- **The input control is missing from Figure 4E.**

The input control has been presented as a supplementary figure EV9 in our revised manuscript (page 9).

- **An empty virus control is missing from the experiments presented in Figure 5C-D.**

Thank you for this comment. As a control virus, we were using Ad-RFP virus at the same MOI as Ad-hSETX for each biological repeat, however Ad-RFP appeared to be toxic to cells and no images could be taken and analysed for that condition. The data presented in Figure 5C, D were generated from embryonic motor neurons extracted from the SMNdelta7 mouse model. Unfortunately, we are unable to repeat this experiment with an empty virus control due to breeding problems that we are facing with our SMNdelta7 colony. Due to the COVID-19 pandemic and the lockdown which began in March 2020, then the shift working system, our SMNdelta7 colony was all but lost. Upon attempting to re-establish the colony by purchasing new Het pairs we have faced significant issues obtaining KO animals, meaning we are currently unable to isolate embryonic motor neurons from these animals. We will have to purchase further pairs to re-establish this mouse colony which could take several months. However, we are confident with the validity of our results for the following reasons; firstly, according to Bernabo et al., translational impairment is a cell-autonomous event in SMA that is regulated directly by SMN levels (Bernabo, Tebaldi et al. 2017), which is exactly what we detect after transduction of SMA motor neurons with LV-SMN vector where the translational impairment is reversed. Secondly, viral-mediated overexpression of both SMN and SETX leads to improvement of translational defects in SMN-deficient motor neurons even though two different viral vector systems were utilised (lentivirus and adenovirus, respectively).

- **The data presented in Figure 6B are variable and would benefit from an additional experiment or two to confirm the reported phenotype.**

We sought some advice from two independent biostatisticians on how to present the data highlighted in Figure 6B. Following their advice, we normalised each of the three data sets (three independent biological replicates). To do this, we calculated the average value in each biological replicate of all the UT points (normalizer). Then, we divided all the points (UT vs CPT) by that UT average value for each replicate. Finally, we plotted all the normalized points from all three replicates in a single graph and performed a statistical test. Since UT and CPT data sets were not normally distributed (as tested with Shapiro-Wilk test), we used a non-parametric test called Mann-Whitney test. The data are now presented in the revised Figure 6B after incorporating the recommendations from the biostatisticians (page 10).

- **SMN protein levels in all of the models presented in Figure 1 should really be shown, like they have for the iPSCs.**

A supplementary figure was included (Figure EV6) showing SMN protein levels in SMA embryonic motor neurons, spinal cords isolated from SMA pre-symptomatic mice (P2) and SMA fibroblasts (page 6 of the revised manuscript)

- **Red/green immunofluorescent colour-combinations should be avoided (e.g. Fig. 2) due to issues for certain forms of colour-blindness. Please see the following link and amend the figures accordingly: <https://www.ascb.org/science-news/how-to-make-scientific-figures-accessible-to-readers-with-color-blindness/>**

Thank you for highlighting this point. Figure 2 has been altered accordingly

- **Please check citations of the EV figures within the text; e.g., EV5 on page 8, should be EV6.**

The citations have been altered accordingly

- **Unlike the other transfected SMN proteins, SMN E134K looks to be excluded from the nucleus (Figure EV6) - could this explain the reduced interaction with RNA Pol?**

The low magnification images that we presented in Figure EV6 of initial submission may have not allowed the real appreciation of the SMN E134K localisation. For this reason, we included higher magnification images of all our constructs and a more representative image of SMN E134K, in particular (Figure EV8). Similar to what has been reported previously (Morse, Shaw et al. 2007, Sabra, Texier et al. 2013), SMN harbouring the E134K point mutation is indeed targeted to the nucleus. Therefore, its disrupted interaction with RNA polymerase I cannot be attributed to its absence from the nucleus.

- **The labelling/statistics on Figure 5D are unclear.**

The following sentence has been added in Figure 5 legend: 'Data is presented as mean \pm s.e.m. # $P < 0.05$ (comparing SMA cells treated with LV-SMN and SMN untreated cells), ## $P < 0.01$ (comparing SMA cells treated with Ad-SETX and SMA untreated cells) and ** $P < 0.01$ (comparing SMA and control untreated cells) (see revised manuscript Page 35).

- **Please provide all antibody catalogue numbers in the methods.**

The antibody catalogue numbers has been included in the immunocytochemistry section of materials and methods (page 19): The primary antibodies utilised in this study included SMN (1:1000, BD, 610646), γ H2AX (1:1000, Merk Millipore, 05-636), DNA/RNA hybrids [S9.6] (Boguslawski et al

1986) (1:2000, Kerafast), Nucleolin (1:2000, Abcam, ab22758), DDX21 (1:500, Novus Biologicals, NB100-1718).

• Please include the age of SMN delta 7 mice used in the main text and Figure legends.

The age of SMNdelta7 mice used (postnatal day 2) has also been indicated in the main text (page 5).

• Please check all scale bars - e.g., Figure 1C: both cannot be 10 µm.

The correct scale bars have been added (Figure 1C of initial submission is now Figure EV2). The following sentence has been included in the Figure Legends section of supplementary figures “Scale bars represent 50 µm (left image) and 10 µm (right image), respectively.

• Please provide details of any reference genes and primer sequences used in the qPCR experiments.

The reference genes and primer sequences used in the qPCR experiments have been included in the RT-qPCR section of materials and methods (page 24).

mouse GAPDH F (reference gene)	CAACTTTGGTATCGTGGAAGGAC
mouse GAPDH R (reference gene)	ACAGTCTTCTGGGTGGCAGTG

Response to Reviewer 3

• Authors show that DDX21 is selectively downregulated in motor neurons and its deficiency may contribute to R-loop accumulation, DNA damage and neurodegeneration associated with SMA. (a) Authors should examine the effect of siRNA-based knockdown of DDX21 in iPSC-derived motor neurons from controls.

A new figure (Figure 9) presenting data obtained upon DDX21 knockdown in iPS-derived motor neurons isolated from three healthy individuals was generated. The manuscript has also been altered accordingly: “To investigate the impact of DDX21 deficiency in motor neuron rDNA integrity, we knocked down DDX21 in iPS-MNs isolated from three healthy individuals. Depletion of DDX21 led to increased nucleolar γ H2AX levels in those neurons (Figure 9 A-L). This finding suggests that the motor neuron-specific deficiency of DDX21 in SMA may exacerbate the rDNA damage phenotype seen in SMN-deficient motor neurons and may contribute to their neurodegeneration.” (Page 12) and “...and rDNA damage, thus impacting on protein synthesis. Knockdown of DDX21 in healthy iPS-MNs led to increased nucleolar γ H2AX levels, reinforcing our hypothesis.” (Page 15). A figure legend was also included (pages 37-38). Methods details included in page 19.

- **The effect of complementation using overexpression of recombinant DDX21 (adeno or lentiviral vectors) in SMA motor neurons on the rescue to R-loops and degenerative phenotype.**

This is indeed a very interesting experiment that would determine whether DDX21 replacement could rescue the neurodegenerative phenotype seen in SMA. However, such an experiment needs careful consideration and meticulous design given the unwanted and potentially pathogenic effect of DDX21 overexpression. DDX21 is overexpressed in human breast and colon cancer tissues. It is also upregulated in 25% of human neuroblastoma. Therefore, it is apparent that the dosage of DDX21 protein in the target cells is concerning and a careful balance of protein expression is required that will allow the assessment of the therapeutic potential of DDX21 without activating other pathways that could lead to pathology. In an attempt to address this comment we generated an LV-DDX21 vector, and empty LV control. In our hands, overexpression of DDX21 in control iPS-MNs (via lentiviral-mediated delivery), resulted in increased pan-nuclear γ H2AX staining (characteristic of cells in S phase). This was accompanied by alterations in cellular morphology including loss of iPS-MNs neuronal processes and enlarged soma, reminiscent of a fibroblast-like morphology. These could be signs of post-mitotic cells re-entering the cell cycle. For this reason we were unable to assess the effect of DDX21 overexpression on the rescue of SMA degenerative phenotypes.

As per the editorial recommendation, we focused our efforts on the impact of DDX21 depletion. The data are reported in Figure 9 of the revised manuscript.

- **There are inconsistencies reported in the literature about the use of S9.6 antibody in immunofluorescence-based quantification of R-loop because S9.6 antibody interacts with dsRNA. I recommend using an additional method for R-loop quantitation, a dot blot method using purified genomic DNA to strengthen the finding (Ramirez, P. et al, J. Vis. Exp., 2021; Grunseich, C. et al., Mol. Cell, 2018).**

To begin with, it is important to highlight that R-loop-mediated DNA damage is widely reported in SMA (Zhao, Gish et al. 2016, Jangi, Fleet et al. 2017, Kannan, Jiang et al. 2020). However, we do agree that due to the inconsistencies mentioned above, more than one method to quantify R-loops is required. It was for this reason that we performed orthogonal assays such as HB-GFP ChIP-qPCR in order to further back-up our results obtained through S9.6 staining. We believe that this method is more accurate, and relevant for this study, compared to dot blot as it specifically focuses on R-loops formed on ribosomal DNA. As reviewed in Chedin, Harton et al. 2021, which outlines the best practises for the visualisation and mapping of R-loops, because dot blots measure total R-loop loads, including mitochondrial R-loops or RNA/DNA hybrids forming at repetitive regions such as telomeres, they are not highly sensitive and are difficult to quantify precisely. For this reason, dot blots may mask or skew the more subtle changes in R-loops over distinct genomic regions, and caution

is advised when interpreting their results (Chedin, Hartono et al. 2021). With this in mind we feel that dot-blot analysis of R-loops would not significantly strengthen the findings of this manuscript.

• **Discussion section, last paragraph. Please expand and clarify the statement "Alternatively, DDX21 may play a more direct role in ribosome biology by reducing rRNA levels upstream of translation."**

The following sentence was added in page 14 of the main text: It has been reported that DDX21 directly binds to rRNA and snoRNAs, promoting rRNA transcription and processing (Calo et al 2015). Reduced levels of DDX21 in motor neurons may lead to insufficient binding of DDX21 to rDNA loci resulting in inhibition of rRNA transcription.

References

Bernabo, P., T. Tebaldi, E. J. N. Groen, F. M. Lane, E. Perenthaler, F. Mattedi, H. J. Newbery, H. Zhou, P. Zuccotti, V. Potrich, H. K. Shorrock, F. Muntoni, A. Quattrone, T. H. Gillingwater and G. Viero (2017). "In Vivo Translatome Profiling in Spinal Muscular Atrophy Reveals a Role for SMN Protein in Ribosome Biology." Cell Rep **21**(4): 953-965.

Chedin, F., S. R. Hartono, L. A. Sanz and V. Vanoosthuyse (2021). "Best practices for the visualization, mapping, and manipulation of R-loops." EMBO J **40**(4): e106394.

Morse, R., D. J. Shaw, A. G. Todd and P. J. Young (2007). "Targeting of SMN to Cajal bodies is mediated by self-association." Hum Mol Genet **16**(19): 2349-2358.

Renvoise, B., G. Querol, E. R. Verrier, P. Bulet and S. Lefebvre (2012). "A role for protein phosphatase PP1gamma in SMN complex formation and subnuclear localization to Cajal bodies." J Cell Sci **125**(Pt 12): 2862-2874.

Sabra, M., P. Texier, J. El Maalouf and P. Lomonte (2013). "The Tudor protein survival motor neuron (SMN) is a chromatin-binding protein that interacts with methylated lysine 79 of histone H3." Journal of Cell Science **126**(16): 3664-3677.

El Hage, A., S. L. French, A. L. Beyer and D. Tollervey (2010). "Loss of Topoisomerase I leads to R-loop-mediated transcriptional blocks during ribosomal RNA synthesis." Genes Dev **24**(14): 1546-1558.

Ginno, P. A., P. L. Lott, H. C. Christensen, I. Korf and F. Chedin (2012). "R-loop formation is a distinctive characteristic of unmethylated human CpG island promoters." Mol Cell **45**(6): 814-825.

Jangi, M., C. Fleet, P. Cullen, S. V. Gupta, S. Mekhoubad, E. Chiao, N. Allaire, C. F. Bennett, F. Rigo, A. R. Krainer, J. A. Hurt, J. P. Carulli and J. F. Staropoli (2017). "SMN deficiency in severe models of spinal muscular atrophy causes widespread intron retention and DNA damage." Proc Natl Acad Sci U S A **114**(12): E2347-e2356.

Jurga, M., A. A. Abugable, A. S. H. Goldman and S. F. El-Khamisy (2021). "USP11 controls R-loops by regulating senataxin proteostasis." Nat Commun **12**(1): 5156.

Kannan, A., X. Jiang, L. He, S. Ahmad and L. Gangwani (2020). "ZPR1 prevents R-loop accumulation, upregulates SMN2 expression and rescues spinal muscular atrophy." Brain **143**(1): 69-93.

Massé, E., P. Phoenix and M. Drolet (1997). "DNA topoisomerases regulate R-loop formation during transcription of the *rrnB* operon in *Escherichia coli*." J Biol Chem **272**(19): 12816-12823.

Nadel, J., R. Athanasiadou, C. Lemetre, N. A. Wijetunga, Ó. B. P. H. Sato, Z. Zhang, J. Jeddeloh, C. Montagna, A. Golden, C. Seoighe and J. M. Grealley (2015). "RNA:DNA hybrids in the human genome have distinctive nucleotide characteristics, chromatin composition, and transcriptional relationships." Epigenetics Chromatin **8**: 46.

Sollier, J., C. T. Stork, M. L. García-Rubio, R. D. Paulsen, A. Aguilera and K. A. Cimprich (2014). "Transcription-coupled nucleotide excision repair factors promote R-loop-induced genome instability." Mol Cell **56**(6): 777-785.

Walker, C., S. Herranz-Martin, E. Karyka, C. Liao, K. Lewis, W. Elsayed, V. Lukashchuk, S. C. Chiang, S. Ray, P. J. Mulcahy, M. Jurga, I. Tsagakis, T. Iannitti, J. Chandran, I. Coldicott, K. J. De Vos, M. K. Hassan, A. Higginbottom, P. J. Shaw, G. M. Hautbergue, M. Azzouz and S. F. El-Khamisy (2017). "C9orf72 expansion disrupts ATM-mediated chromosomal break repair." Nat Neurosci **20**(9): 1225-1235.

Zhao, D. Y., G. Gish, U. Braunschweig, Y. Li, Z. Ni, F. W. Schmitges, G. Zhong, K. Liu, W. Li, J. Moffat, M. Vedadi, J. Min, T. J. Pawson, B. J. Blencowe and J. F. Greenblatt (2016). "SMN and symmetric arginine dimethylation of RNA polymerase II C-terminal domain control termination." Nature **529**(7584): 48-53.

February 16, 2022

Re: Life Science Alliance manuscript #LSA-2021-01145-TR

Prof. Mimoun Azzouz
University of Sheffield
Neuroscience
385a Glossop Rd, Broomhall
Sheffield S10 2HQ
United Kingdom

Dear Dr. Azzouz,

Thank you for submitting your revised manuscript entitled "SMN-deficient cells exhibit increased ribosomal DNA damage" to Life Science Alliance. The manuscript has been seen by the original reviewers whose comments are appended below. While the reviewers continue to be overall positive about the work in terms of its suitability for Life Science Alliance, some important issues raised by Rev 3 remain. Specifically, please perform the staining with S9.6 antibody in cell treated with siDDX21 to confirm that DNA damage caused by knockdown of DDX21 (siDDX21) is R-loop mediated.

Our general policy is that papers are considered through only one revision cycle; however, given that the suggested changes are relatively minor, we are open to one additional short round of revision. Please note that I will expect to make a final decision without additional reviewer input upon resubmission.

Please submit the final revision within one month, along with a letter that includes a point by point response to the remaining reviewer comments.

To upload the revised version of your manuscript, please log in to your account: <https://lsa.msubmit.net/cgi-bin/main.plex>
You will be guided to complete the submission of your revised manuscript and to fill in all necessary information.

B. MANUSCRIPT ORGANIZATION AND FORMATTING:

Sincerely,

Reviewer #1 (Comments to the Authors (Required)):

The authors have done an excellent job at constructively responding to all of my concerns on the initially submitted manuscript. I am happy for this work to be published in its current form.

Reviewer #3 (Comments to the Authors (Required)):

Authors have performed some experiments and improved the manuscript. However, authors have argued not to perform dot-blot analysis for R-loops, which is a bit strange. Complementation experiment did not work in iPSC MN and resulted in an unexpected phenotype, which could be interesting. Authors could have simply tested any other established cell line for complementation experiment. The following are some suggestions to further improve and strength this manuscript.

Fig. 9. Why authors have not included data on staining with S9.6 antibody in cell treated with siDDX21? Only staining for γ H2AX is shown. It is very important to know that the DNA damage caused by knockdown of DDX21 (siDDX21) is R-loop mediated. I suggest including S9.6 staining data.

Many places in figures γ H2AX labelled as γ H2AX. Replace "y" with Greek letter for "gamma".

Authors should update citations as new papers have been published on the role of DNA damage and R-loops in SMA in the last 6 months.

RE: Second revision of manuscript LSA-2021-01145-T

Karyka et al.

Reviewer #1

The authors have done an excellent job at constructively responding to all of my concerns on the initially submitted manuscript. I am happy for this work to be published in its current form.

We are so pleased to see that Reviewer 1 was satisfied by our efforts and appreciated the value of our new data.

Reviewer #3

Authors have performed some experiments and improved the manuscript. However, authors have argued not to perform dot-blot analysis for R-loops, which is a bit strange. Complementation experiment did not work in iPSC MN and resulted in an unexpected phenotype, which could be interesting. Authors could have simply tested any other established cell line for complementation experiment.

We appreciate the constructive comments of Reviewer 3. Regarding the dot-blot analysis for R-loops, we explained in detail in our previous letter our reservations with this assay and the reasons why we chose to prioritise other experiments. However, given that we have had time to perform second revisions of our manuscript we have now performed dot-blot analysis of R-loops in HeLa cells treated with DDX21 targeted siRNA (data added as Fig S10). In line with the previously published work (Song et al 2017), knockdown of DDX21 led to a significant increase in R-loops detected by dot-blot (Fig S10D and E).

As for the complementation experiment, our preliminary data were very interesting, indeed. Since DDX21 protein appears to be specifically decreased in SMA motor neurons (Fig 7D and K) and not in SMA cortical neurons (Fig 8D) or SMA fibroblasts (Fig 8H), we argue that the complementation experiment should be performed in motor neurons. However, as we mentioned before, we are currently facing some serious breeding problems with our SMNdelta7 colony, therefore it was impossible for us to use murine embryonic SMA motor neurons for this experiment. Our only alternative was to use human iPSC-MNs. We are confident that the overexpression experiment worked in its technical aspects because we observed increased levels of DDX21 protein after lentiviral (LV) transduction compared to transduction with empty lentiviral vector as a control. Furthermore, the observed phenotypes after LV-DDX21 vector transduction (e.g., increased pan-nuclear γ H2AX staining that is characteristic of cells in S-phase, alterations in cellular morphology, enlarged soma, etc., which are signs of cells re-entering the cell cycle) were in line with the current bibliography and the link between upregulation of DDX21 and cancer (Jung et al 2011, Putra et al 2021, Zhang et al 2014).

The following are some suggestions to further improve and strengthen this manuscript.

1. Fig. 9. Why authors have not included data on staining with S9.6 antibody in cell treated with siDDX21? Only staining for γ H2AX is shown. It is very important to know that the DNA damage caused by knockdown of DDX21 (siDDX21) is R-loop mediated. I suggest including S9.6 staining data.

We agree with Reviewer 3 that it is important to demonstrate the DNA damage caused by loss of DDX21 is R-loop mediated. In line with this request we have now included Supplementary Figure 10 (Fig S10) showing that, consistent with previously reported data (Song et al 2017), siRNA targeted knock down of DDX21 leads to increased nuclear/nucleolar S9.6 (R-loop) staining compared to control siRNA treated cells (Fig S10A, B and C). These immunofluorescent data are also supported by the findings of the S9.6 dot-blot analysis (Fig S10D and E). The DDX21 antibody is incompatible with the methanol:acetone fixation method which is required for S9.6 staining. Therefore, DDX21 knockdown was confirmed by western blot analysis on the same batches of cells used in these assays (Fig S10F and G)

In Fig 9 a dual staining of γ H2AX and DDX21 was used to correlate γ H2AX signal intensity with confirmed DDX21 knocked down cells. Given the relatively low efficiency of Accell siRNA knock down in iPSC-MNs, a dual staining of DDX21 and S9.6 would therefore also be crucial for this R-loop experiment. However, correlating S9.6 signal intensity with DDX21 knocked down cells would not have been possible given the methanol:acetone fixation method used. We therefore chose to perform these siRNA experiments in HeLa cells where we can achieve extremely efficient knockdown, as demonstrated in Fig S10F and G.

Given that DDX21 protein is a nucleolar helicase, the fact that increased R-loops formation upon DDX21 knockdown has been previously reported (Song et al 2017) and our new data presented in Fig S10, we are therefore confident that the increased nucleolar DNA damage we observed in Fig.9 after DDX21 knockdown is indeed R-loop mediated.

2. Many places in figures γ H2AX labelled as γ H2AX. Replace "y" with Greek letter for "gamma".

Figures 2, 6 and 9 have been altered accordingly. Similar changes have been made on pages 6 and 11 of the revised manuscript.

3. Authors should update citations as new papers have been published on the role of DNA damage and R-loops in SMA in the last 6 months.

Citations have been updated, accordingly. In particular, three citations that had previously been missed out were included in the Discussion section (Page 13). We appreciate Reviewer 3 bringing this to our attention.

- Song C, Hotz-Wagenblatt A, Voit R, Grummt I. 2017. SIRT7 and the DEAD-box helicase DDX21 cooperate to resolve genomic R loops and safeguard genome stability. *Genes Dev* 31: 1370-81
- Jung Y, Lee S, Choi HS, Kim SN, Lee E, et al. 2011. Clinical validation of colorectal cancer biomarkers identified from bioinformatics analysis of public expression data. *Clinical cancer research : an official journal of the American Association for Cancer Research* 17: 700-9
- Putra V, Hulme AJ, Tee AE, Sun JQJ, Atmadibrata B, et al. 2021. The RNA-helicase DDX21 upregulates CEP55 expression and promotes neuroblastoma. *Molecular oncology* 15: 1162-79
- Zhang Y, Baysac KC, Yee LF, Saporita AJ, Weber JD. 2014. Elevated DDX21 regulates c-Jun activity and rRNA processing in human breast cancers. *Breast cancer research : BCR* 16: 449

March 31, 2022

RE: Life Science Alliance Manuscript #LSA-2021-01145-TRR

Prof. Mimoun Azzouz
University of Sheffield
Neuroscience
385a Glossop Rd, Broomhall
Sheffield S10 2HQ
United Kingdom

Dear Dr. Azzouz,

Thank you for submitting your revised manuscript entitled "SMN-deficient cells exhibit increased ribosomal DNA damage". We would be happy to publish your paper in Life Science Alliance pending final revisions necessary to meet our formatting guidelines.

- please add the Twitter handle of your host institute/organization as well as your own or/and one of the authors in our system
- please add your supplementary figure legends to the main manuscript text, directly after the main figure legends
- tables should be included at the bottom of the main manuscript file or be sent as separate files
- we encourage you to revise the figure legends for figures 7 and S7 such that the figure panels are introduced in alphabetical order
- please add callouts for Figures 3A, 4D, S1A-B, S6A-D, S7A-C to your main manuscript text;
- please indicate scale bar sizes in Legend for figure S7

A. FINAL FILES:

B. MANUSCRIPT ORGANIZATION AND FORMATTING:

Sincerely,

April 4, 2022

RE: Life Science Alliance Manuscript #LSA-2021-01145-TRRR

Prof. Mimoun Azzouz
University of Sheffield
Neuroscience
385a Glossop Rd, Broomhall
Sheffield S10 2HQ
United Kingdom

Dear Dr. Azzouz,

Thank you for submitting your Research Article entitled "SMN-deficient cells exhibit increased ribosomal DNA damage". It is a pleasure to let you know that your manuscript is now accepted for publication in Life Science Alliance. Congratulations on this interesting work.

DISTRIBUTION OF MATERIALS:

Again, congratulations on a very nice paper. I hope you found the review process to be constructive and are pleased with how the manuscript was handled editorially. We look forward to future exciting submissions from your lab.

Sincerely,
